# Luqin-like RYamide peptides regulate food-evoked responses in *C. elegans*

**Hayao Ohno[1†], Morikatsu Yoshida[2], Takahiro Sato[3], Johji Kato[4], Mikiya Miyazato[2], Masayasu Kojima[3], Takanori Ida[4,5]\*, Yuichi Iino[1]\***

[1]Department of Biological Sciences, Graduate School of Science, The University of Tokyo, Tokyo, Japan; [2]Department of Biochemistry, National Cerebral and Cardiovascular Center Research Institute, Osaka, Japan; [3]Molecular Genetics, Institute of Life Sciences, Kurume University, Fukuoka, Japan; [4]Department of Bioactive Peptides, Frontier Science Research Center, University of Miyazaki, Miyazaki, Japan; [5]Center for Animal Disease Control, University of Miyazaki, Miyazaki, Japan

**Abstract** Peptide signaling controls many processes involving coordinated actions of multiple organs, such as hormone-mediated appetite regulation. However, the extent to which the mode of action of peptide signaling is conserved in different animals is largely unknown, because many peptides and receptors remain orphan and many undiscovered peptides still exist. Here, we identify two novel *Caenorhabditis elegans* neuropeptides, LURY-1-1 and LURY-1-2, as endogenous ligands for the neuropeptide receptor-22 (NPR-22). Both peptides derive from the same precursor that is orthologous to invertebrate luqin/arginine-tyrosine-NH$_2$ (RYamide) proneuropeptides. LURY-1 peptides are secreted from two classes of pharyngeal neurons and control food-related processes: feeding, lifespan, egg-laying, and locomotory behavior. We propose that LURY-1 peptides transmit food signals to NPR-22 expressed in feeding pacemaker neurons and a serotonergic neuron. Our results identified a critical role for luqin-like RYamides in feeding-related processes and suggested that peptide-mediated negative feedback is important for satiety regulation in *C. elegans*.

DOI: https://doi.org/10.7554/eLife.28877.001

**\*For correspondence:**
a0d203u@cc.miyazaki-u.ac.jp (TI);
iino@bs.s.u-tokyo.ac.jp (YI)

**Present address:** [†]Developmental Biology Program, Sloan Kettering Institute, New York, United States

**Competing interests:** The authors declare that no competing interests exist.

## Introduction

Deciphering the mechanisms of appetite control and the physiological effects of feeding is of great significance in modern societies, in which eating disorders and obesity are major health issues. The regulation of feeding requires interactions between distant organs and flexible fine-tuning of neural circuitries; therefore, it is no wonder that many neuromodulators including peptide hormones have been reported to control feeding in animals (*Coll et al., 2007*; *Sobrino Crespo et al., 2014*; *Marder, 2012*). Previous studies in mammals have suggested the importance of negative feedback mediated by peptide hormones in satiety signaling. For example, feeding induces the secretion of anorexigenic (feeding-inhibitory) hormones such as peptide YY (PYY), pancreatic polypeptide (PP), cholecystokinin (CCK), and glucagon like peptide-1 (GLP-1) from the digestive organs, which in turn prevent excessive feeding (*Perry and Wang, 2012*). These peptides counteract or negatively regulate neuropeptide Y (NPY), which is orexigenic (feeding-stimulatory). Although a wealth of knowledge has been obtained in mammalian models for food-related function of these peptides, understanding the roles of each peptide is not straightforward because of the multitude of cells and tissues involved and the multiplicate regulations acting on the peptides. Extensive studies have also been conducted on feeding and gastric motor circuits in invertebrates. For example, in *Aplysia*, the buccal ganglion feeding circuit is known to be regulated by neuropeptides such as *Aplysia* NPY,

allatotropin-related peptide (ATRP), and myomodulin gene 2-derived peptide (MMG2-DP), while the feeding-related circuits in the stomatogastric ganglion in crustaceans are also regulated by known neuropeptides, as well as classical neurotransmitters (*Marder, 2012*; *Taghert and Nitabach, 2012*). It is however still unclear to what extent these regulatory logics are universal and commonly inherited in the animal kingdom.

The nematode *Caenorhabditis elegans* exhibits a variety of food-related behavioral and physiological plasticity despite its simple body structure (*Ashrafi, 2007*; *Avery and You, 2012*; *Hart, 2006*). *C. elegans* feeds on bacteria and ingests them by a pumping and peristaltic movements of the pharynx (or foregut) (*Avery and You, 2012*). The pumping rate of the pharynx is finely regulated by food availability and other environmental conditions, while food availability also affects other behaviors such as egg-laying and locomotion (*Avery and You, 2012*; *Waggoner et al., 2000*; *You et al., 2008*). For example, when starved *C. elegans* worms abruptly encounter food, pharyngeal pumping and egg-laying become highly active (*Dong et al., 2000*; *You et al., 2008*). Such enhanced food intake during the refeeding period then induces a cessation of feeding and locomotion, which is thought to result from satiety (*You et al., 2008*). This detailed knowledge of food-related behaviors, along with the small number of cells and the simplicity of body structure, makes *C. elegans* a useful model system to investigate the evolutionarily conserved regulatory mechanisms of feeding and satiety. Monoamine neurotransmitters and some of the arginine-phenylalanine-NH$_2$ (RFamide) neuropeptides are known to be involved in food-related regulations and in some cases the exact cells in which these signaling molecules act have been identified (*Chase and Koelle, 2007*; *Cohen et al., 2009*; *Li and Kim, 2008*; *Papaioannou et al., 2005*; *Waggoner et al., 2000*). Previous studies suggest that peptidergic signaling often acts as a component of a feedback loop and fine-tunes food-related neuronal circuits (*Chalasani et al., 2010*; *Cohen et al., 2009*).

Understanding the mechanisms of feeding control in nematodes such as *C. elegans* is also industrially important, because feeding is vital and can be a promising target process for pest control. For example, ivermectin, which is one of the most successful drugs both in human and livestock medicine (*Omura and Crump, 2004*), has been considered to exert its nematocidal effects on *C. elegans* through the constitutive opening of AVR-14/AVR-15-containing glutamate-gated chloride channels, and thus prevents feeding (*Keane and Avery, 2003*).

Our current understanding of feeding control by peptide signaling in animals is still largely limited, because (1) many peptides and peptide receptors are still orphan (i.e. for which a binding partner(s) has yet to be identified), (2) feeding is a very complicated process that involves most organs in the body, (3) the relative genetic intractability of the mammalian system makes it difficult to determine the in vivo functions of peptides and their receptors, and (4) the search for novel peptides and comparative analyses of peptides across classes or species have been hindered by their small size and low sequence conservation.

The mammalian NPY peptide family, which is composed of NPY, PYY, and PP, is characterized by sequence features such as the C-terminal arginine-tyrosine-NH$_2$ (RYamide) motif and their involvement in appetite control (*Loh et al., 2015*; *Tatemoto, 1982*). Five receptors for the NPY peptide family have been cloned from mammals (Y1, Y2, Y4, Y5, and Y6). These receptors belong to the superfamily of seven-transmembrane G protein-coupled receptors (GPCR). NPY family peptides perform a variety of physiological functions by activating different subtype receptors (*Loh et al., 2015*).

In invertebrates, peptides containing the C-terminal RYamide structure were first discovered in the brachyuran crab *Cancer borealis* (*Li et al., 2003*) and are also found in diverse classes, such as crustaceans (*Christie, 2014*; *Dircksen et al., 2011*; *Ma et al., 2010*), mollusks (*Proekt et al., 2005*; *Veenstra, 2010*), and insects (*Hauser et al., 2010*; *Ida et al., 2011*; *Roller et al., 2016*). Some RYamide-containing peptides have been previously implied to have a role in feeding; the abundance of FVGGSRYamide and SGFYANRYamide increases after food intake in *Cancer borealis* (*Chen et al., 2010*), QPPLPRYamide (MMG2-DPb) modulates contractions of a feeding-related muscle in *Aplysia californica* (*Proekt et al., 2005*), and the gene encoding EERTDMSPVPFVMGSRYamide and SPARLI TPRNDRFFMGSRYamide is expressed in the enteroendocrine cells of the midgut in *Bombyx mori* (*Roller et al., 2016*).

Luqins, a family of peptides originally isolated from left upper quadrant (LUQ) neurons in *Aplysia*, are peptides with around 10 amino acids with C-terminal RFamide, RYamide, or RWamide, and also characterized by a motif containing two cysteines located in the C-terminal portion of their precursors (*Jékely, 2013*; *Mirabeau and Joly, 2013*; *Semmens et al., 2016*; *Zatylny-Gaudin and Favrel,*

2014). Some members of precursors that encode a RYamide-containing peptide(s) in ecdysozoa are considered to be closely orthologous to lophotrochozoan luqin precursors, and this family of ecdysozoan peptides has been historically called RYamides (*Jékely, 2013*; *Mirabeau and Joly, 2013*). Orthologs of luqin/RYamides precursors have been also found in deuterostomes (*S. purpuratus* and *S. kowalevskii*), but not in chordates (*Jékely, 2013*; *Mirabeau and Joly, 2013*).

Using nucleic acid-based homology screening approaches, a *Drosophila melanogaster* GPCR, CG5811 (also known as NepYR), was identified and found to be activated by mammalian NPY and PYY (*Li et al., 1992*). Endogenous ligands for CG5811 had long been unknown. Recently, we identified two RYamide-containing peptides in *Drosophila melanogaster*, named dRYamide-1 and -2 (*Ida et al., 2011*), as potent endogenous ligands for CG5811 (*Collin et al., 2011*; *Ida et al., 2011*) (Figure 2C). dRYamides modulate feeding behavior in the blowfly *Phormia regina* (*Ida et al., 2011*; *Maeda et al., 2015*). However, little is known about the precise physiological roles for dRYamides and CG5811 in regulation of feeding and other behaviors. In *C. elegans*, FMRFamide-like peptide-7 (FLP-7) derived peptides were reported to be ligands for neuropeptide receptor-22 (NPR-22), which is phylogenetically similar to CG5811 (www.wormbase.org; *Figure 1—figure supplement 1*; *Mertens et al., 2006*; *Palamiuc et al., 2017*). This phylogenic analysis shows that NPR-22 and CG5811 belong to the luqin/RYamide receptor family and may be distantly related to tachykinin, NPY, and leucokinin receptors (*Figure 1—figure supplement 1*), although the evolutionary relationship between the luqin/RYamide receptor group and other receptor groups has not so far been clear (*Jékely, 2013*; *Mirabeau and Joly, 2013*). FLP-7 precursor does not have characteristics of luqins, such as a canonical pair of cysteines in the C-terminal portion, and all of the peptides generated contain C-terminal RFamide structures, but not RYamide structures. Therefore, it is still largely elusive how the repertoire and the mode of action of luqins and RYamides are conserved in *C. elegans*.

Here, we report the identification of two luqin-like RYamide peptides in *C. elegans*, named LURY-1-1 and -2, as endogenous ligands for NPR-22. Importantly, LURY-1 peptides are secreted from two classes of pharyngeal neurons in a food-dependent manner and regulate multiple food-related processes including feeding and egg-laying through NPR-22. LURY-1 peptides act on the feeding pacemaker neurons MC and the serotonergic RIH neuron to exert their functions. Our results propose that luqin-like RYamide peptides are important for satiety transmission.

## Results

### Identification and structural determination of LURY-1-1 and LURY-1-2

To identify novel peptide ligands in *C. elegans*, we constructed a stable CHO cell line expressing NPR-22a (CHO-NPR-22a) and measured changes in intracellular calcium concentration ($[Ca^{2+}]_i$) induced by gel filtration samples from worm extracts. However, we could not detect any agonist activity from each fraction in this $Ca^{2+}$ assay (*Figure 1A*). In contrast to the NPR-22a-expressing cells, our previous study showed that CG5811-expressing cells have a distinctive feature to detect both RYamide- and RFamide-containing peptides with high potency (*Ida et al., 2011*). Therefore, we utilized a stable CHO cell line expressing CG5811 (CHO-CG5811) to search for agonist peptides from *C. elegans* extracts. The agonist activity for CHO-CG5811 cells was revealed at molecular mass (*Mr*) of about 1000 by gel filtration chromatography (*Figure 1A*). This activity could not be detected on CHO-NPR-22a cells even by the use of ten times the amount of samples used on CHO-CG5811 cells (*Figure 1A*). When the active fractions were separated by carboxymethyl ion-exchange high-performance liquid chromatography (CM-HPLC) at pH 6.5, four distinct agonist activities were revealed (fractions A–D, *Figure 1B*). Each activity was purified as a single peak by successive reverse-phase HPLC (RP-HPLC). Of these, two peaks, P1 and P2, were isolated from fraction A (*Figure 1—figure supplement 2A,B*). Amino acid sequences of the purified peptides were determined by a protein sequencer and mass spectrometry (*Table 1*). These two peptides were encoded by the same gene, *Y75B8A.11*, which was predicted as an ortholog of luqin/RYamide precursors in a bioinformatic study (*Mirabeau and Joly, 2013*). Combined with the sequence of the *Y75B8A.11* cDNA (*Figure 2A,B*), we deduced the primary structures of the peptides as AVLPRY-NH$_2$ (P1) and PALLSRY-NH$_2$ (P2) (*Figure 2C*). Mass spectrometric analysis revealed that the observed monoisotopic *m/z* values of the purified peptides (P1, 717.25 and P2, 818.34) were almost identical to the theoretically predicted values (717.42 and 818.46, respectively) for peptides that are C-terminally amidated. Moreover, the

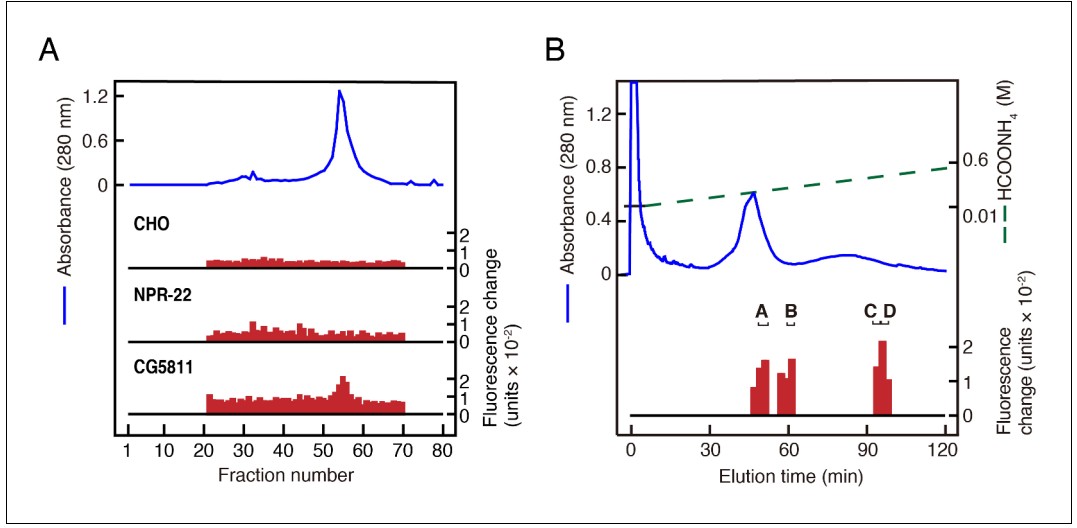

**Figure 1.** Purification of LURY-1 peptides. (**A**) Gel filtration on Sephadex G-50 (fine) of the SP-III fraction obtained from *C. elegans*. Red bars indicate fluorescence changes owing to [$Ca^{2+}$]$_i$ increase in receptor expressing cells. (top) Non-transfected CHO cells. (middle) CHO-NPR-22a cells. (bottom) CHO-CG5811 cells. The amount of worm extracts in each [$Ca^{2+}$]$_i$ assay was approximately equivalent to that from 20 mg (top, middle) or 2 mg (bottom) frozen weight of *C. elegans*. (**B**) CM-HPLC (pH 6.5) of the active fractions on the gel filtration. Red bars indicate [$Ca^{2+}$]$_i$ changes in CHO-CG5811 cells. Active fractions that were further fractionated are depicted as A to D. (**A, B**) Blue lines indicate total protein content in each fraction as measured by absorbance at 280 nm.

DOI: https://doi.org/10.7554/eLife.28877.002

The following figure supplements are available for figure 1:

**Figure supplement 1.** Phylogenetic analysis of *C. elegans* NPR-22 with neuropeptide GPCRs.
DOI: https://doi.org/10.7554/eLife.28877.003

**Figure supplement 2.** Identification of LURY-1 peptides.
DOI: https://doi.org/10.7554/eLife.28877.004

synthetic AVLPRY-NH$_2$ and PALLSRY-NH$_2$ had identical retention times to natural peptides on RP-HPLC (*Figure 1—figure supplement 2C*). These data also suggest that both of the natural peptides have the deduced primary structures. These peptides have the C-terminal RYamide motif identical to that of dRYamide peptides, the authentic ligands for CG5811 (*Figure 2C*; *Ida et al., 2011*). We assigned *Y75B8A.11* the gene name *lury-1* (LUqin-like RYamide peptides) and designated the PALLSRY-NH$_2$ and AVLPRY-NH$_2$ peptides as LURY-1-1 and LURY1-2, respectively. The proneuropeptide has a characteristic of luqins with canonical pair of cysteines in the C-terminal portion (*Figure 2B*). Each of the peptides purified from fractions B to D was determined to be an RFamide-

**Table 1.** Active fractions of each chromatography and the amino acid sequences of the purified peptides.

| Ion-exchange-HPLC Elution time (min) | | Reversed-phase-HPLC Retention time (min) | Observed mass (MH$^+$) | Theoretical mass (MH$^+$) | Sequence | Precursor gene | Peptide name* |
|---|---|---|---|---|---|---|---|
| pH 6.5 | pH4.8 | | | | | | |
| 48–52 (Fr. A) | 46–48 | 13–13.5 | 717.25 | 717.42 | AVLPRY-NH$_2$ | *lury-1* | LURY-1-2 |
| | | 17–18 | 818.34 | 818.46 | PALLSRY-NH$_2$ | *lury-1* | LURY-1-1 |
| 60–62 (Fr. B) | 56–60 | 25.5–26.5 | 792.29 | 792.43 | PNFLRF-NH$_2$ | *flp-1* | FLP-1-6 |
| 92–94 (Fr. C) | 108–112 | 27.5–28.5 | 1255.45 | 1255.68 | RNKFEFIRF-NH$_2$ | *flp-12* | FLP-12 |
| 94–96 (Fr. D) | 86–88 | 15.5–16.5 | 901.27 | 901.45 | KSAYMRF-NH$_2$ | *flp-6* | FLP-6-1 |
| | 90–92 | 23–23.5 | 1108.33 | 1108.55 | SPSAKWMRF-NH$_2$ | *flp-22* | FLP-22 |

*Peptide names other than LURY-1 peptides are taken from **Li and Kim, 2008**.
DOI: https://doi.org/10.7554/eLife.28877.007

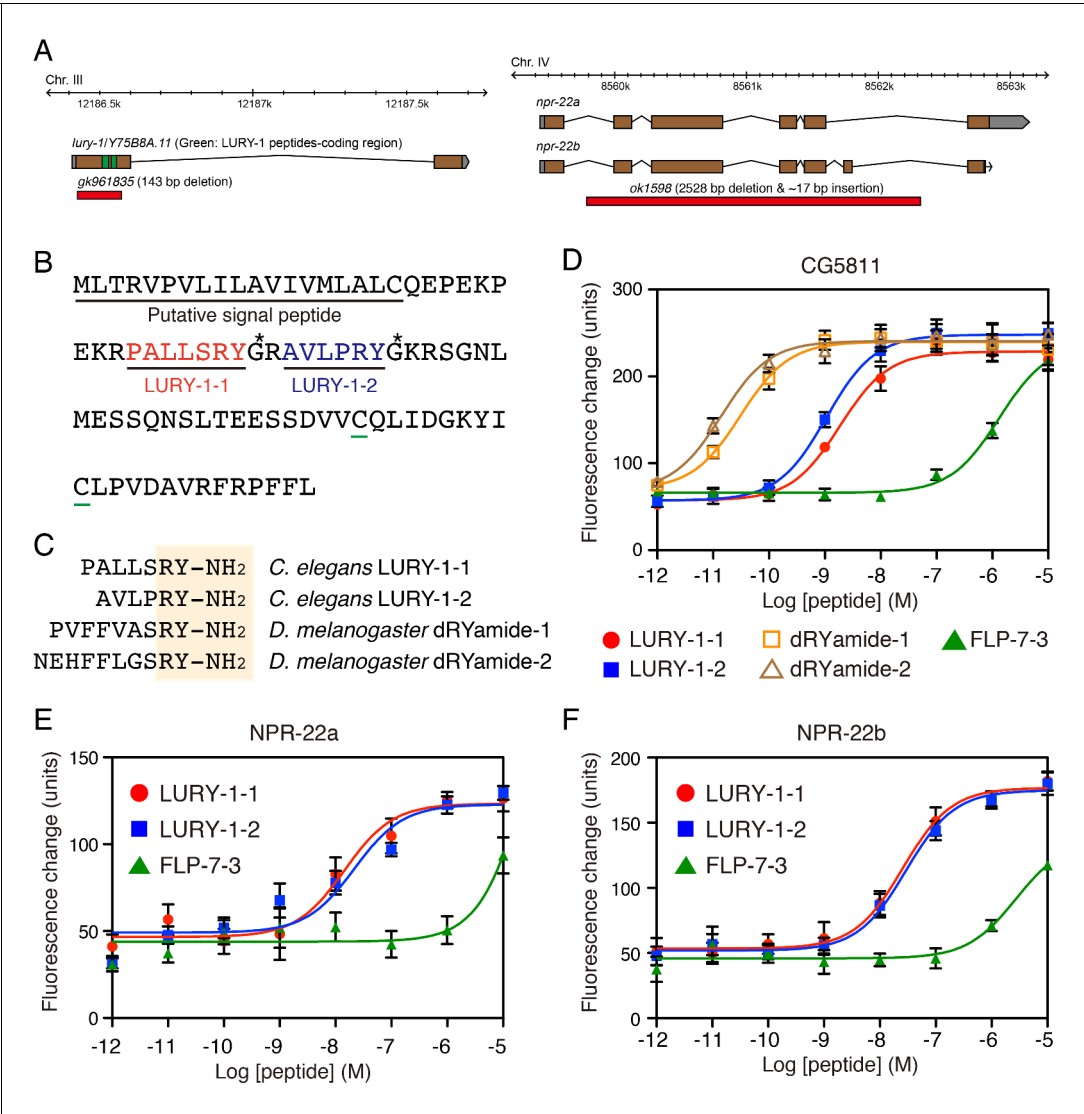

**Figure 2.** LURY-1 peptides are potent ligands for NPR-22. (**A**) Genomic organization of the *lury-1/Y75B8A.11* (left) and *npr-22* (right) loci and the lesions in mutant alleles. (**B**) Amino acid sequences of LURY-1-1 (red) and LURY-1-2 (blue) in the LURY-1 prepropeptide. *lury-1* encodes an 89-residue peptide. The asterisks show glycine residues, which serve as amide donors for C-terminal amidation. The green underlines show cysteine residues which are conserved among propeptides of the luqin family. (**C**) Sequences comparison of *C. elegans* LURY-1 peptides and *Drosophila* dRYamides. RYamide structures conserved between the peptides are shaded. (**D**) Pharmacological characterization of synthetic peptides using CG5811 stably expressed in CHO cells. Dose-response relationships of changes in $[Ca^{2+}]_i$ for LURY-1-1 (red), LURY-1-2 (blue), dRYamide-1 (orange), dRYamide-2 (brown), and FLP-7-3 (green) in CHO-CG5811 cells. (**E, F**) Pharmacological characterization of synthetic peptides using the A isoform (**E**) and the B isoform (**F**) of *C. elegans* NPR-22 stably expressed in CHO cells. Dose-response relationships of changes in $[Ca^{2+}]_i$ for LURY-1-1 (red), LURY-1-2 (blue), and FLP-7-3 (green) in CHO-NPR-22a/b cells. Data points are means ±S.E.M. of 6 replicates for each experiment.

DOI: https://doi.org/10.7554/eLife.28877.005

The following figure supplement is available for figure 2:

**Figure supplement 1.** Pharmacological characterization of synthetic peptides.

DOI: https://doi.org/10.7554/eLife.28877.006

containing peptide generated from *flp-1*, *flp-12*, *flp-6*, or *flp-22* based on protein sequencing and mass spectrometry (*Table 1*). LURY-1-1 and -2 induced concentration-dependent, robust increases in $[Ca^{2+}]_i$ in CHO-CG5811 cells, with a half-maximal response concentration ($EC_{50}$) of $1.86 \times 10^{-9}$ and $1.07 \times 10^{-9}$ M, respectively (*Figure 2D*). Neither LURY-1 peptides nor dRYamides induced a response in CHO cells transfected with vector alone (*Figure 2—figure supplement 1A,B*), confirming that these peptides act through CG5811.

We next investigated whether LURY-1 peptides could bind and activate NPR-22. LURY-1-1 and -2 induced concentration-dependent robust increases in $[Ca^{2+}]_i$ in CHO-NPR-22a cells ($EC_{50} = 1.44 \times 10^{-8}$ and $2.18 \times 10^{-8}$ M, respectively) (*Figure 2E*). Potencies of these peptides for CHO-NPR-22a cells were ~20 fold less than CHO-CG5811 cells. Mertens et al. reported that FLP-7-3 (SPMERSAMVRF-NH$_2$) derived from FLP-7 was the most potent ligand for NPR-22 (*Mertens et al., 2006*). However, the potency of FLP-7-3 for CHO-NPR-22a cells in the $Ca^{2+}$ assay was lower than that of LURY-1 peptides ($EC_{50}$; submicromolar vs. subnanomolar range) (*Figure 2E*). Furthermore, each RFamide-containing peptide purified from fractions B to D had no effect in the assay used for CHO-NPR-22a cells (*Figure 2—figure supplement 1C*). When we used a stable CHO cell line expressing the B isoform of NPR-22 (CHO-NPR-22b), which is the only known splice variant of NPR-22a (*Figure 2A*), LURY-1 peptides showed similar efficacies and potencies to those for the CHO-NPR-22a cells (*Figure 2F*). Based on these findings, we propose that LURY-1-1 and -2, derived from the *lury-1* gene, are endogenous ligands for NPR-22.

## The *lury-1* gene is expressed in two classes of pharyngeal neurons

We next investigated the in vivo expression patterns of the *lury-1* and *npr-22* genes. When the fluorescent reporter Venus was fused to the 3.5 kb *lury-1* promoter (*lury-1^prom^*), the expression of Venus was observed only in two classes of pharyngeal neurons (*Figure 3—figure supplement 1A*). These neurons were identified as the M1 and M2 neurons (*Figure 3A*) by their morphology, position, and the co-expression of established cell markers (*Kim and Li, 2004; Refai et al., 2013*) (*Figure 3—figure supplement 1B,C*). We found that the *lury-1^prom^::Venus* reporter was robustly expressed in M1 in all larval stages and in adults, whereas its expression in M2 was not detectable in most L1–L3 worms but started to increase from the L4 stage (*Figure 3A* and *Figure 3—figure supplement 1D*). We also generated a *lury-1^prom^::lury-1::Venus* translational reporter, in which Venus is C-terminally fused in frame to the genomic *lury-1*-coding sequences containing introns. This translational reporter produced fluorescent signals in coelomocytes, which are macrophage-like scavenger cells that endocytose proteins secreted into the body cavity (*Fares and Greenwald, 2001*), as well as in the M1 and M2 neurons (*Figure 3—figure supplement 1E*). This result implies that the LURY-1 preproprotein is sorted into the secretory pathway and that LURY-1 peptides can reach distant cells.

The Venus reporter driven by the 5.7 kb *npr-22* promoter was expressed in many cells mainly in the head region (*Figure 3B* and *Figure 3—figure supplement 2A*); the *npr-22^prom^::Venus* expression was observed in head muscles, the I2 neurons, the MC neurons, the RIH neuron, the AIA neurons, the AUA neurons, the ASK neurons

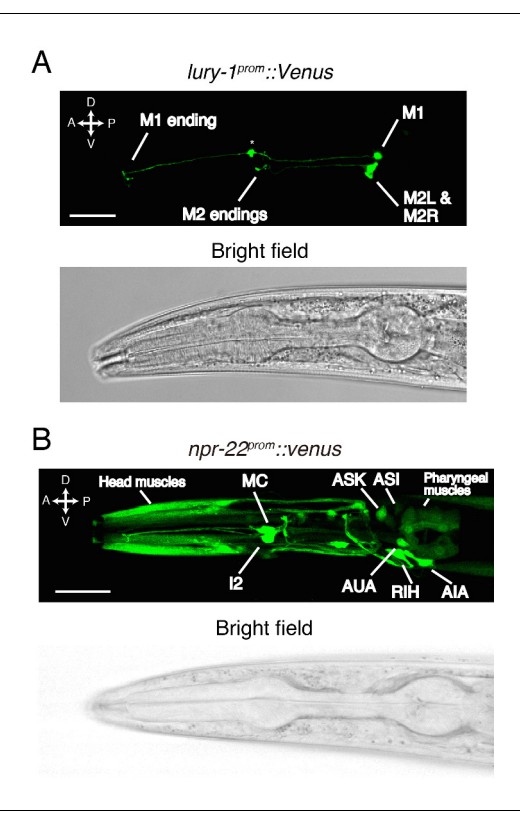

**Figure 3.** LURY-1 peptides are expressed in the pharyngeal M1 and M2 neurons, while NPR-22 is broadly expressed. (**A**) Expression pattern of *lury-1^prom^::Venus* reporter in the head region of an adult worm. Fluorescence of *lury-1^prom^::venus* was specifically observed in the M1 and M2 pharyngeal neurons in adult worms. Asterisk, a knot-like structure in the M1 axon occasionally generated by the expression of fluorescent proteins. (**B**) Expression pattern of Venus reporter driven by the *npr-22* promoter in the head region of an adult worm. (**A, B**) White scale bars, 20 μm.

DOI: https://doi.org/10.7554/eLife.28877.008

The following figure supplements are available for figure 3:

**Figure supplement 1.** Expression patterns of *lury-1*.
DOI: https://doi.org/10.7554/eLife.28877.009

**Figure supplement 2.** Expression patterns of *npr-22*.
DOI: https://doi.org/10.7554/eLife.28877.010

(strong in larval worms), the ASI neurons (strong in larval worms), a few B-type motorneurons in the posterior ventral nerve cord (variable), pharyngeal muscles, body wall muscles (weak), the intestine (weak), and a few classes of unidentified cells anterior to the nerve ring (*Figure 3B* and *Figure 3—figure supplement 2A*). In the I2, MC, RIH, AIA, AUA, and posterior B-type motorneurons, we confirmed the co-expression of established cell markers (*Feng and Hope, 2013*; *Hao et al., 2001*; *Kim and Li, 2004*; *Macosko et al., 2009*; *Sze et al., 2002*; *Tomioka et al., 2006*) (*Figure 3—figure supplement 2B–H*). Collectively, these results raise the possibility that pharyngeal LURY-1 peptides may transmit signals to a broad range of cells.

## LURY-1 peptides regulate egg-laying, feeding, lifespan, and locomotion through their receptor NPR-22

To assess the physiological functions of LURY-1 peptides, we introduced the multicopy array containing a 5.7 kb fragment spanning the *lury-1* locus (extrachromosomal *Ex[lury-1(+)]* or chromosomally integrated *peIs2413[lury-1(+)]/peIs2414[lury-1(+)]*) and examined the effects of the increased copy numbers of *lury-1*. This multicopy expression of *lury-1* caused the following phenotypes. First, numbers of unlaid eggs in the uterus were decreased (*Figure 4A,B*). The rate of egg-laying was not affected by the multicopy *lury-1* expression (*Figure 4C*), indicating that in this strain the rate of ovulation is normal while egg-laying is constitutively facilitated and early-stage embryos are prematurely laid (*Figure 4A*). These phenotypes meet the standard criteria to be classified as egg-laying constitutive (Egl-c) (*Hart, 2006*) and are similar to those of the mutants previously described as Egl-c (*Hawasli et al., 2004*; *Kwok et al., 2006*; *Ringstad and Horvitz, 2008*) (*Figure 4—figure supplement 1A,B*). Second, the rate of pharyngeal pumping, which is a rhythmic muscle contraction required for food intake (*Avery and You, 2012*), was reduced (*Figure 4D*). Third, adult lifespan is extended as much as 21–50% (*Figure 4E*, *Figure 4—figure supplement 2*, and *Supplementary file 1*). The ectopic expression of *lury-1* in all neurons using the *rimb-1* promoter also caused the similar phenotypes (*Figure 4B,D* and *Supplementary file 1*). Fourth, roaming behavior, which was measured by the numbers of grid squares that were entered by the worm tracks (*Fujiwara et al., 2002*), was attenuated (*Figure 4F*).

Importantly, these Egl-c, slow-pumping, long-lived, and reduced-roaming phenotypes were all largely suppressed by the deletion of *npr-22* (*Figure 4A,B,D,E,F*, *Figure 4—figure supplement 2*, and *Supplementary file 1*), indicating that *lury-1(+)* functions upstream of *npr-22*.

To examine whether LURY-1 peptides are responsible for the effects of the multicopy expression of the *lury-1* gene, we next administered synthetic LURY-1 peptides into adult worms using a peptide microinjection protocol (*Rogers et al., 2003*). When a 10 µM solution of either LURY-1-1 or LURY-1-2 was injected into wild-type worms, number of eggs retained in the uterus and pharyngeal pumping rate were reduced (*Figure 5A,B*), similar to worms carrying the *lury-1(+)* transgenic array (*Figure 4B,D*). These effects of the injection of LURY-1 peptides were not observed in the *npr-22 (ok1598)* mutants (*Figure 5A,B*). On the other hand, neither the administration of a 10 µM solution of another putative NPR-22 ligand, FLP-73, nor the multicopy expression of its precursor gene *flp-7 (+)* caused any Egl-c or slow-pumping phenotype (*Figure 5—figure supplement 1A–D*). Taken together, these results support that both LURY-1-1 and LURY-1-2 act through NPR-22 to control multiple processes.

## LURY-1 peptides may transmit food signals

When a proneuropeptide is tagged with a fluorescent protein, the soluble fluorescent protein is considered to be packaged into dense core vesicles (DCVs) together with the processed active neuropeptide(s) and the fluorescence can be used to monitor DCV release (*Sasidharan et al., 2012*; *Sieburth et al., 2007*). In *C. elegans*, the exocytosis of fluorescent proteins can be assessed by quantifying the fluorescence in coelomocytes, into which the proteins in the body cavity are concentrated (*Fares and Greenwald, 2001*). To examine under which conditions LURY-1 peptides are secreted, we performed this coelomocyte uptake assay, which has been used to evaluate the secretion of neuropeptides in previous studies (*Ch'ng et al., 2008*; *Hao et al., 2012*; *Palamiuc et al., 2017*; *Rabinowitch et al., 2016*; *Sasidharan et al., 2012*; *Schild et al., 2014*; *Sieburth et al., 2007*). When Venus is C-terminally fused to the LURY-1 coding region, the fluorescence of the Venus reporter in coelomocytes was decreased in starved worms, and was rapidly restored by subsequent

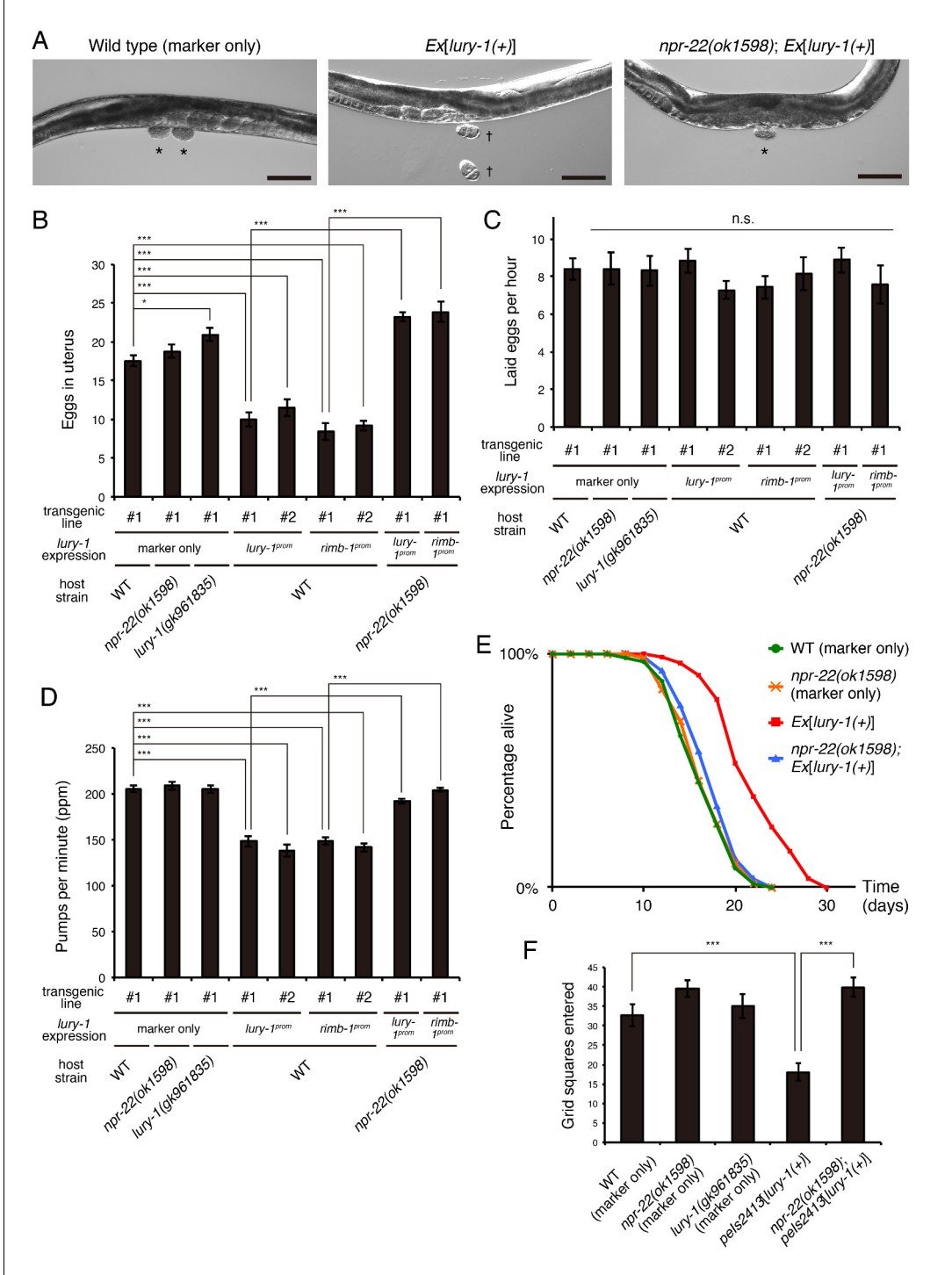

**Figure 4.** Multicopy expression of *lury-1(+)* causes multiple food-related phenotypes in an NPR-22-dependent manner. (A) Lateral views near the uterus of synchronized adult worms (36 hr after the L4 white-crescent stage) of wild type, *Ex[lury-1(+)]*, and *npr-22(ok1598); Ex[lury-1(+)]*. Asterisk, ~30 cell stage eggs laid on agar pads during immobilization. Daggers, 2- to 4 cell stage eggs laid on an agar pad during immobilization. The number of eggs retained in the uterus is decreased in the *Ex[lury-1(+)]* strain. Black scale bars, 100 μm. (B) Number of unlaid eggs in the uterus of *lury-1-* and *npr-22-* related strains. N = 29, 25, 30, 24, 21, 18, 18, 24, and 18 animals (from left to right in order of each group). ***p<0.001, *p<0.05 (ANOVA with Tukey's post-test). (C) Number of eggs laid per animal per hour. N = 8, 8, 8, 8, 8, 7, 7, 8, and 7 animals (from left to right in order of each group). ANOVA with Tukey's post-test. n.s., not significant. (D) Pharyngeal pumping rate of *lury-1-* and *npr-22-* related strains. N = 25, 25, 26, 27, 17, 26, 17, 26, and 25 animals (from left to right in order of each group). ***p<0.001 (ANOVA with Tukey's post-test). (E) Lifespans of *lury-1-* and *npr-22-* related strains. Details are shown in **Supplementary file 1**. (F) Roaming behavior of *lury-1-* and *npr-22-* related strains. Tracks of worm movement generated on food over a 4

*Figure 4 continued on next page*

*Figure 4 continued*

hr period were examined. *N* = 12 animals. ***p<0.001 (ANOVA with Tukey's post-test). (A–F) All strains have the *myo-3^prom^::venus* transgene as a transgenic marker. Bars represent mean ±S.E.M.

DOI: https://doi.org/10.7554/eLife.28877.011

The following figure supplements are available for figure 4:

**Figure supplement 1.** Egg-laying phenotypes of canonical Egl-c mutants.

DOI: https://doi.org/10.7554/eLife.28877.012

**Figure supplement 2.** Multicopy expression of LURY-1 extends lifespan in an NPR-22-depndent manner.

DOI: https://doi.org/10.7554/eLife.28877.013

refeeding (*Figure 6A,B*). When GFP is fused with a signal sequence for secretion and expressed in body wall muscles, the fluorescence is also accumulated in coelomocytes (*Fares and Greenwald, 2001*) (*Figure 6—figure supplement 1A*). This fluorescence reporting the constitutive secretion from muscles was unchanged by starvation or subsequent refeeding (*Figure 6—figure supplement 1B*). Moreover, when Venus alone was expressed under the *lury-1* promoter, the fluorescence in M1 and M2 was unaffected by the presence or absence of food (*Figure 6—figure supplement 1C*). Therefore, the changes in the fluorescence derived from the *lury-1^prom^::lury-1::Venus* reporter in coelomocytes (*Figure 6A,B*) were not likely caused by the modification of the activity of coelomocytes or the *lury-1* promoter. Collectively, these results suggest that the secretion of LURY-1 peptides is positively controlled by the presence of food.

The deletion of *lury-1* or *npr-22* (*Figure 2A*) by itself did not cause egg-laying defective, fast-pumping, short-lived, or increased-roaming phenotype under normal conditions (*Figure 4B,D,E,F* and *Supplementary file 1*), except for a weak egg-laying defective phenotype shown by the *lury-1 (gk961835)* mutants (*Figure 4B*). These observations suggested the possibility that LURY-1 peptides exert their effects only under specific conditions. Considering that *lury-1* is expressed in the pharyngeal M1 and M2 neurons, LURY-1 peptides might transmit the changes in the pharyngeal activity.

To address this possibility, we examined the egg-laying and pharyngeal pumping behavior during refeeding after fasting, the period during which pharyngeal pumping is highly activated and subsequently suppressed (*You et al., 2008*) (*Figure 6E*, WT). In wild-type worms, the egg-laying behavior is dramatically inhibited upon food deprivation but it resumes if they are returned to food, which apparently increases the offspring's chance of growing in a food-rich environment (*Dong et al., 2000*). We found that the *lury-1(gk961835)* and *npr-22(ok1598)* mutants normally retained eggs in their body after 2 hr starvation (*Figure 6C*). However, in the *lury-1(gk961835)* and *npr-22(ok1598)* mutants, the numbers of eggs laid during the initial 30 min refeeding period were decreased (*Figure 6D*), and the numbers of unlaid eggs in the uterus returned to baseline of fed worms more slowly during the refeeding period than in wild type (*Figure 6C*). We also found that the *lury-1 (gk961835)* and *npr-22(ok1598)* mutants normally elevated their pharyngeal pumping shortly after refeeding, but showed their satiety-induced suppression of pharyngeal pumping (*You et al., 2008*) more slowly than wild type (*Figure 6E*). These results might suggest that the activities of LURY-1 peptides and NPR-22 are latent under normal culture conditions and that LURY-1 peptides exert their actions under certain conditions that induce hyperactivation of pharynx, such as abrupt refeeding after starvation, along with other signals such as insulin and TGF-β (*You et al., 2008*).

## NPR-22 acts in distinct cells to control feeding and egg-laying

To determine where the LURY-1 peptides-mediated signals are received, we expressed the *npr-22* cDNA in the *peIs2413*[*lury-1(+)*]; *npr-22* mutants using various cell-specific promoters and examined whether the effects of multicopy expression of *lury-1(+)* were restored.

When either of the two *npr-22* isoforms (*npr-22a* and *npr-22b*; *Figure 2A*) was expressed under the authentic *npr-22* promoter, both isoforms were found to be able to restore the egg-laying-stimulatory and feeding-suppressive effects of the multicopy expression of *lury-1(+)* in the *npr-22* mutant background (*Figure 7—figure supplement 1A* and *Figure 8—figure supplement 1A*), consistent with the results of the [Ca$^{2+}$]$_i$ assays showing that both isoforms are functional LURY-1 peptides receptors (*Figure 2E,F*). We used *npr-22a* for the following cell-specific rescue experiments.

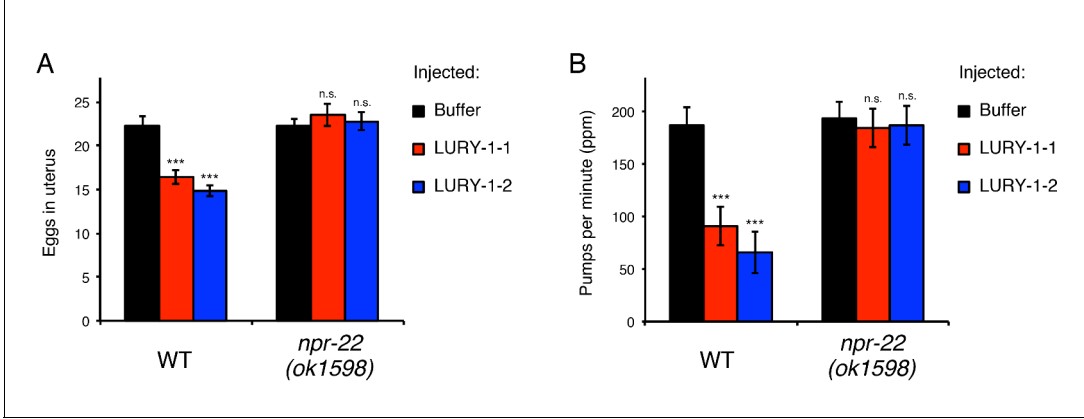

**Figure 5.** LURY-1 peptides stimulate egg-laying and suppress feeding through NPR-22. (A, B) Effects of administration of LURY-1 peptides on the number of retained eggs (A) and pharyngeal pumping (B). Each of synthetic LURY-1 peptides was injected to wild-type and *npr-22(-)* worms at a concentration of 10 μM. M9 dilution buffer was injected as a control. *N* = 16 animals. ***p<0.001 compared with buffer controls (ANOVA with Dunnett's post-test). n.s., not significant. Bars represent mean ±S.E.M.

DOI: https://doi.org/10.7554/eLife.28877.014

The following figure supplement is available for figure 5:

**Figure supplement 1.** Administration of FLP-7–3 does not affect egg-laying or feeding.

DOI: https://doi.org/10.7554/eLife.28877.015

We found that the expression of *npr-22a* driven by the *ceh-19* promoter, which induces the expression in MC, ADF, and PHA (*Feng and Hope, 2013*) (*Figure 3—figure supplement 2C*), was sufficient for the mediation of the suppressive effects of *lury-1(+)* on feeding (*Figure 7A*). In contrast, when *npr-22a* was expressed in *npr-22*-expressing cells other than MC (driven by the *ins-1, ges-1, cat-1, flp-15, myo-2, myo-3,* and *acr-2* promoters), in ADF (driven by the *cat-1* promoter), or in PHA (driven by the *flp-15* promoter), feeding was not affected in the *peIs2413[lury-1(+)]; npr-22* background (*Figure 7A* and *Figure 7—figure supplement 1B*), suggesting that the expression of *npr-22a* in MC is important. To further restrict the expression site of *npr-22* to the left/right pair of the single MC class (MCL and MCR), we next employed the FLP-FRT gene activation system (*Davis et al., 2008*), in which FLP-mediated recombination between two FRT sites allows the transgene expression (*Davis et al., 2008*) (*Figure 7B*). When *npr-22prom::FLP* and *ceh-19prom::FRT::terminator::FRT (FTF)::npr-22a::sl2::mCherry* transgenes were introduced into worms, the mCherry transcriptional reporter placed downstream of the trans-spliced leader sequence SL2 was specifically expressed in the two MC neurons (*Figure 7C*), in which both *npr-22prom* and *ceh-19prom* are active (*Figure 3—figure supplement 2C*). The MC-specific expression of *npr-22a* using this system was sufficient for the mediation of the suppressive effects of *lury-1(+)* on feeding in the *npr-22* mutant background (*Figure 7D*). Therefore, we conclude that NPR-22 is likely to act in MC, which are known to be important neurons for the stimulation of food pumping (*Raizen et al., 1995; Trojanowski et al., 2014*), to control feeding. The expression of *npr-22a* driven by the *ceh-19* promoter also rescued the shortened lifespan of *peIs2413[lury-1(+)]; npr-22* (*Figure 7E*), implying the relationship between feeding suppression and extended lifespan mediated by *lury-1(+)*.

In the regulation of egg-laying, the Egl-c phenotype reappeared when *npr-22a* was expressed in serotonergic/dopaminergic neurons under the *cat-1* promoter in the *peIs2413[lury-1(+)]; npr-22* strain (*Figure 8A*). The RIH neuron is the only cell in which the expression patterns induced by the *npr-22* and *cat-1* promoters overlap (*Figure 3—figure supplement 2D*). Moreover, the Egl-c phenotype was also restored by the expression of *npr-22a* under the *slt-1* promoter (*Figure 8A*), which is also active in the RIH neuron (*Hao et al., 2001*) (*Figure 3—figure supplement 2E*). We also observed that the RIH-specific expression of *npr-22a* driven by the FLP-FRT gene activation system using *npr-22prom* and *cat-1prom* (*Figures 7B* and *8B*) rescued the *npr-22* mutant phenotype in the egg-laying regulation (*Figure 8C*), whereas the expression of *npr-22a* in *npr-22*-expressing cells other than RIH had no effects (*Figure 8A,C* and *Figure 8—figure supplement 1B*). These results suggest that the action of NPR-22 in RIH is important for the facilitation of egg-laying.

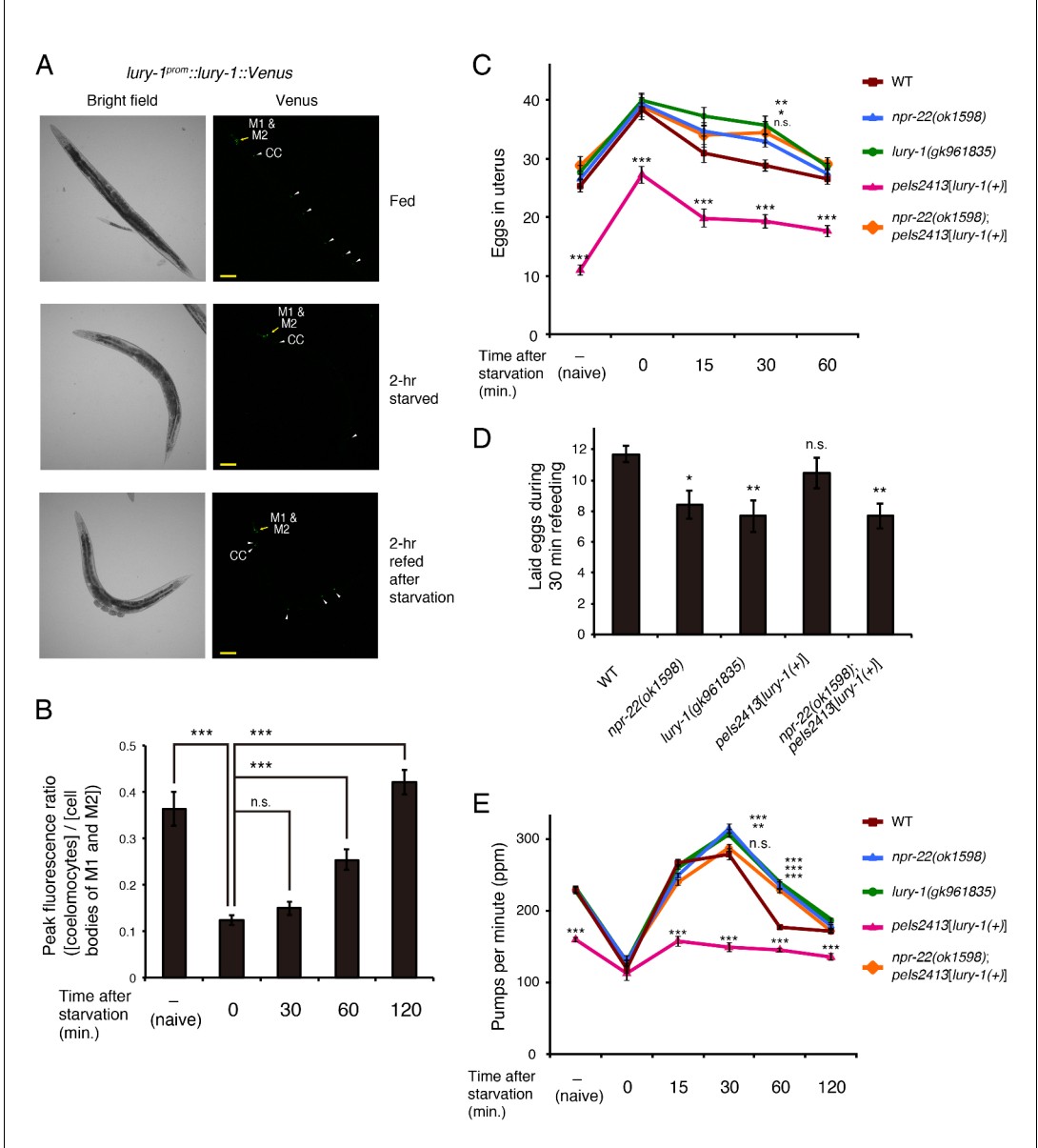

**Figure 6.** LURY-1 peptides are secreted in a food-dependent manner and essential for satiety-induced behavioral changes. (A) The fluorescence of *lury-1^prom^::lury-1::venus* reporter in worms that had been fed (top), fasted (middle), or fasted and refed (bottom). White arrowheads, coelomocytes (CC). Scale bars, 100 μm. (B) The secretion of the LURY-1::Venus reporter, expressed as ratios of fluorescence in coelomocytes to that in the M1 and M2 cell bodies, was quantified in worms that had been fed (naive), starved for two hours, or refed for 30–120 min after the 2 hr starvation. *N* = 15, 20, 16, 16, and 16 animals (from left to right in order of each condition). ***p<0.001 (ANOVA with Dunnett's post-test). (C) Changes in number of unlaid eggs in the uterus. Animals were starved for two hours, and then refed for 15, 30, or 60 min. The results of animals that did not experience starvation are also shown ('naïve'). *N* = 16, 15, 16, 16, and 15 animals (from left to right in order of each condition). ***p<0.001, **p<0.01, *p<0.05 compared with wild-type animals (ANOVA with Dunnett's post-test). (D) Number of eggs laid per animal during 30 min refeeding. After animals were starved for two hours, they were transferred back to fresh *E. coli*-seeded plates. *N* = 16 animals. **p<0.01, *p<0.05 compared with wild-type animals (ANOVA with Dunnett's post-test). (E) Changes in pharyngeal pumping rate. Worms were starved for two hours, and then refed for 15, 30, 60, or 120 min. The results of animals that did not experience starvation are also shown ("naive"). ***p<0.001, **p<0.01 compared with wild-type animals. ns., not significant. Bars represent mean ±S.E.M.

DOI: https://doi.org/10.7554/eLife.28877.016

The following figure supplement is available for figure 6:

**Figure supplement 1.** The activities of coelomocytes and the *lury-1* promoter are unaffected by starvation or refeeding.

DOI: https://doi.org/10.7554/eLife.28877.017

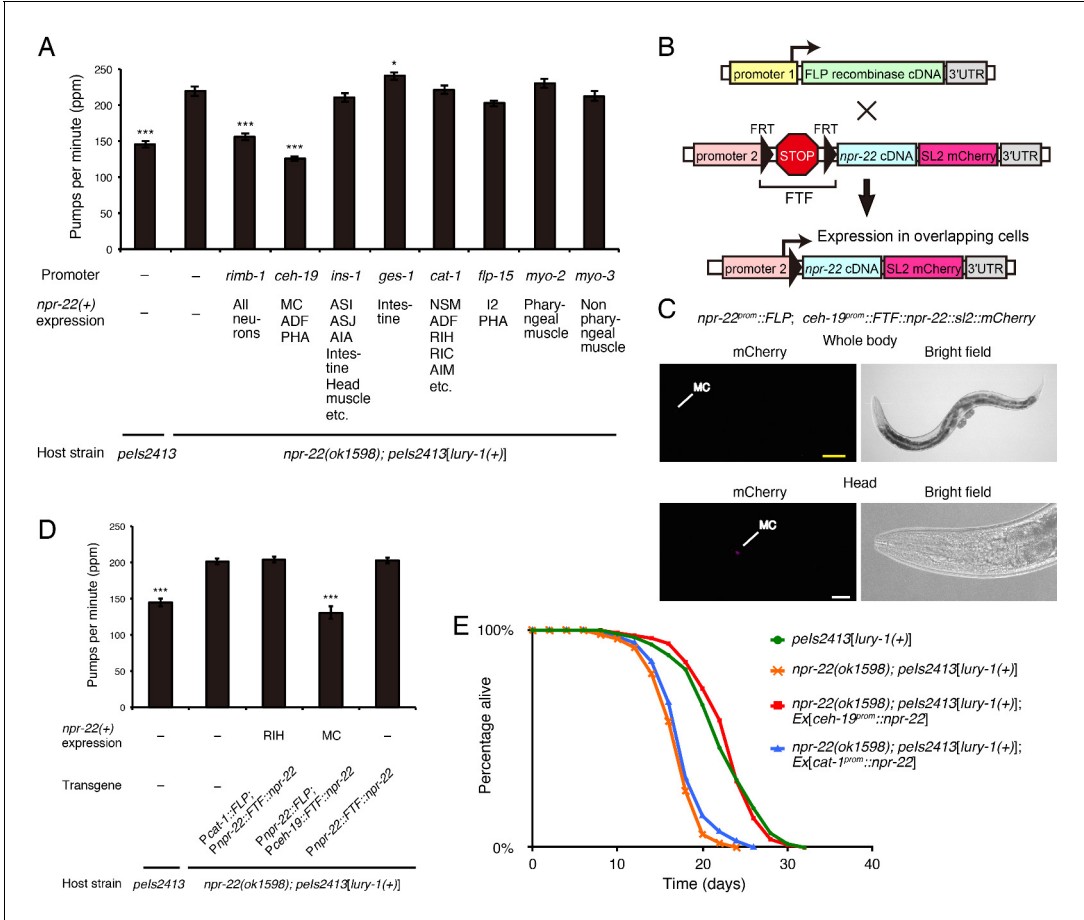

**Figure 7.** NPR-22 acts in MC to control feeding and lifespan. (A) Pharyngeal pumping rate of animals expressing *npr-22a* under indicated promoters. *N* = 13, 12, 12, 12, 12, 12, 12, 12, 12, and 12 animals (from left to right in order of each group). (B) Schematic of the FLP recombinase technology to restrict the *npr-22* expression to single cells. (C) MC-specific expression of the mCherry reporter driven by the combination of *npr-22prom::FLP* and *ceh-19prom::FTF::npr-22::sl2::mCherry* transgenes. The magnified view of the head is shown in the lower panels. Yellow scale bar, 100 μm. White scale bar, 20 μm. (D) Pharyngeal pumping rate of animals expressing *npr-22a* specifically in RIH or MC. *N* = 16 animals. (E) Lifespans of animals expressing *npr-22a* driven by the *ceh-19* or *cat-1* promoter. Details are shown in **Supplementary file 1**. (A, D) ***$p<0.001$, *$p<0.05$ compared with *npr-22(ok1598); peIs2413* animals carrying only the transformation marker (−) (ANOVA with Dunnett's post-test). Bars represent mean ±S.E.M.

DOI: https://doi.org/10.7554/eLife.28877.018

The following figure supplement is available for figure 7:

**Figure supplement 1.** Both NPR-22a and NPR-22b are functional for the regulation of feeding.

DOI: https://doi.org/10.7554/eLife.28877.019

## Serotonin signaling may act downstream of the LURY-1 peptides in egg-laying regulation

The RIH neuron does not express serotonin [5-hydroxytryptamine (5-HT)] biosynthesis enzymes but uptake serotonin from extracellular space through MOD-5/serotonin reuptake transporter (SERT) (*Jafari et al., 2011*), and serotonin regulates many food-associated processes including egg-laying (*Chase and Koelle, 2007*; *Sze et al., 2000*). To examine whether serotonin signaling is involved in the LURY-1 peptides-mediated control of egg-laying, we examined the genetic interaction between *lury-1* and genes for monoaminergic signaling.

The Egl-c phenotype caused by the multicopy expression of *lury-1(+)* was partially suppressed by a nonsense mutation in *cat-1* (*Figure 9A*), which encodes the putative synaptic vesicular transporter for serotonin, dopamine, tyramine, and octopamine (*Chase and Koelle, 2007*; *Duerr et al., 1999*). Moreover, a null mutation in *tph-1*, which encodes the serotonin synthetic enzyme tryptophan

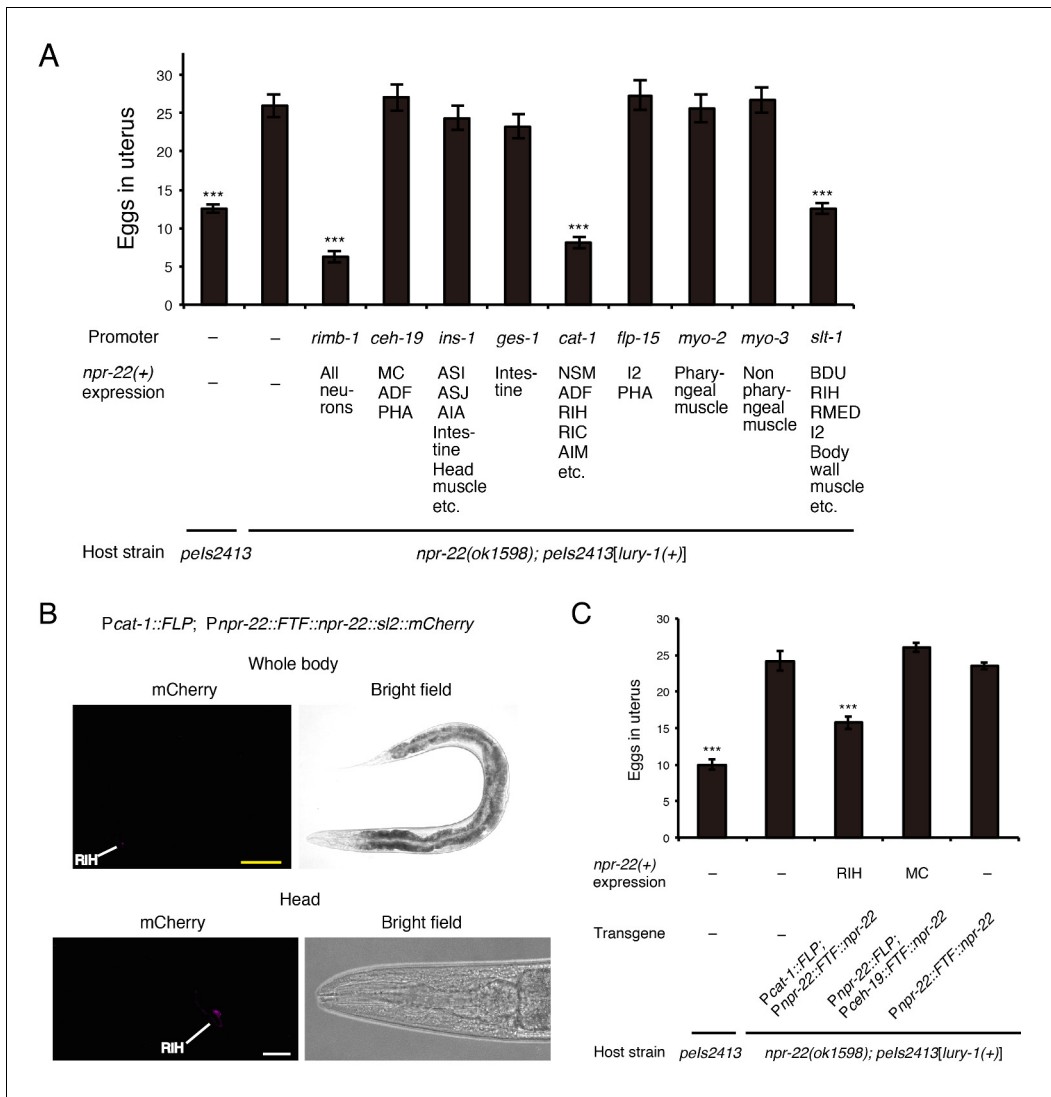

**Figure 8.** NPR-22 acts in RIH to control egg-laying. (**A**) Number of unlaid eggs in the uterus of animals expressing *npr-22a* under indicated promoters. N = 18, 18, 18, 19, 18, 18, 20, 18, 18, 19, and 18 animals (from left to right in order of each group). (**B**) RIH-specific expression of the mCherry reporter driven by the combination of *cat-1^{prom}::FLP* and *npr-22^{prom}::FTF::npr-22::sl2::mCherry* transgenes. The magnified view of the head is shown in the lower panels. Yellow scale bar, 100 μm. White scale bar, 20 μm. (**C**) Number of unlaid eggs in the uterus of animals expressing *npr-22a* specifically in RIH or MC. N = 16 animals. (**A, C**) ***p<0.001 compared with *npr-22(ok1598); peIs2413* animals carrying only the *unc-122^{prom}::mCherry* transformation marker (−) (ANOVA with Dunnett's post-test). Bars represent mean ±S.E.M.

DOI: https://doi.org/10.7554/eLife.28877.020

The following figure supplement is available for figure 8:

**Figure supplement 1.** Both NPR-22a and NPR-22b are functional for the regulation of egg-laying.
DOI: https://doi.org/10.7554/eLife.28877.021

hydroxylase, suppressed the Egl-c phenotype (*Figure 9A*), whereas a null mutation in *cat-2*, which encodes the dopamine synthetic enzyme tyrosine hydroxylase, did not affect egg-laying (*Figure 9A*).

When two alleles of *mod-5* mutants, *mod-5(n3314)* and *mod-5(n822)*, experienced starvation and then refed, the numbers of unlaid eggs decreased more slowly than wild type during the refeeding period (*Figure 9B*), like in the *lury-1* and *npr-22* mutants (*Figure 6C*). Furthermore, the Egl-c phenotype caused by the multicopy *lury-1(+)* expression was partially suppressed by the mutations in *mod-*

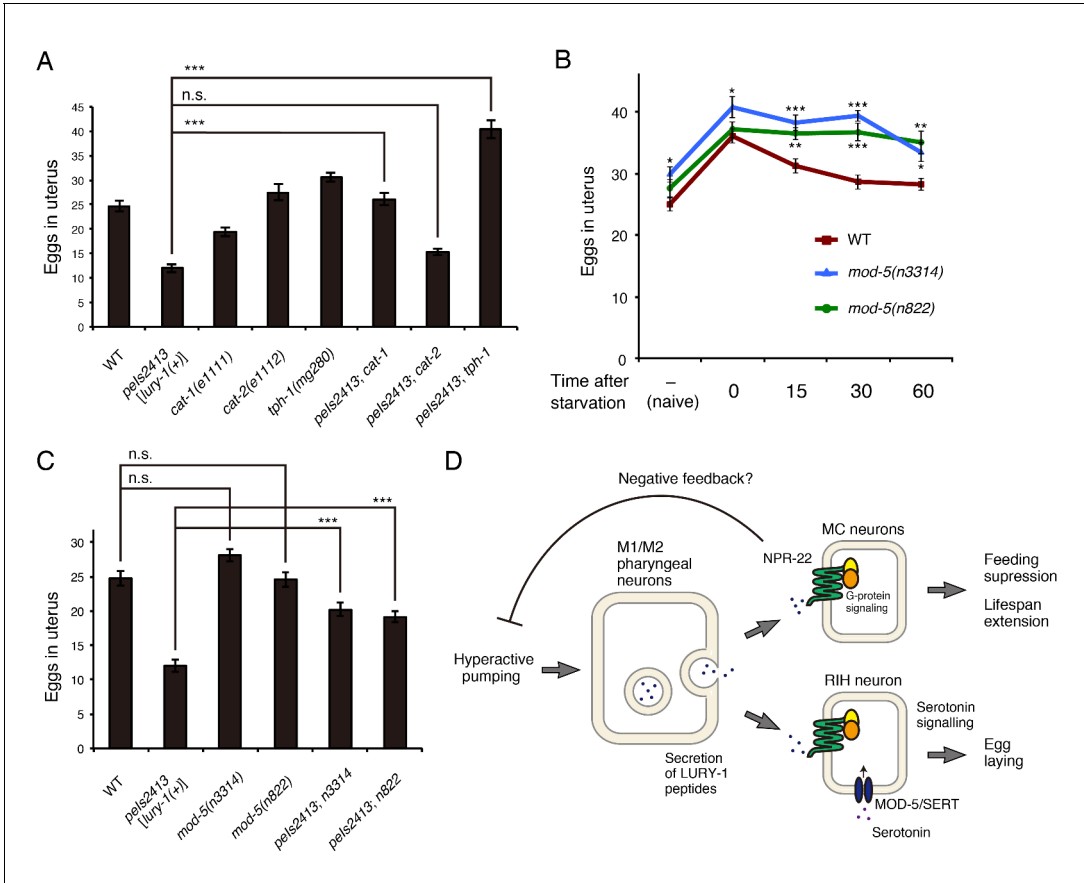

**Figure 9.** Serotonin signaling is essential for the LURY-1 peptide-dependent stimulation of egg-laying. (**A**) Genetic interaction between *peIs2413[lury-1 (+)]* and monoamine-related genes in number of unlaid eggs in the uterus. N = 20 animals. (**B**) Changes in number of unlaid eggs in the uterus. Animals were starved for two hours, and then refed for 15, 30, or 60 min. The results of animals that did not experience starvation are also shown ('naïve'). N = 16 animals for each condition. (**C**) Genetic interaction between *peIs2413[lury-1(+)]* and two alleles of *mod-5* in number of unlaid eggs in the uterus. N = 20 animals. The data for WT and *pe2413* are the same as shown in (**A**). ***p<0.001 (ANOVA with Tukey's post-test). (**D**) Schematic model representing the mechanism of action of LURY-1 peptides. (**A, B**) ***p<0.001, **p<0.01, *p<0.05 (ANOVA with Dunnett's post-test). n.s., not significant. Bars represent mean ±S.E.M.
DOI: https://doi.org/10.7554/eLife.28877.022

5 (*Figure 9C*). Collectively, these results imply that LURY-1 peptides facilitate egg-laying through serotonin signaling derived from RIH.

## Discussion

Using a reverse pharmacological technique, we here identify pharyngeal LURY-1-1 and -2 peptides, encoded by the *lury-1* gene, as endogenous ligands for NPR-22. LURY-1 peptides are secreted from the pharyngeal neurons M1 and M2 and control feeding, egg-laying, lifespan, and roaming behavior through NPR-22. Our cell-specific rescue experiments and genetic analyses suggest that NPR-22 acts in the feeding pacemaker MC neurons to control feeding and lifespan, whereas it acts upstream of serotonergic signals derived from the serotonin-uptaking RIH neuron to control egg-laying. Based on these results, together with the observation that the *lury-1(gk961835)* and *npr-22(ok1598)* deletion mutants reduced egg-laying and elevated feeding during the refeeding period after starvation, we propose a model in which LURY-1 peptides convey information about the food-evoked activation of pharynx to control multiple processes through the broadly expressed NPR-22 (*Figure 9D*). The distance and the lack of synaptic connection between the LURY-1 peptides-secreting neurons and RIH suggest that LURY-1 peptides can function in a hormone-like manner.

Several FMRFamide-related peptides, such as FLP-7–3, have previously been reported to be ligands of NPR-22 (*Mertens et al., 2006*; *Palamiuc et al., 2017*). However, in our hands, FLP-7–3 elicited a much weaker calcium response in CHO-NPR-22 cells than LURY-1 peptides (*Figure 2E,F*) and did not affect egg laying nor pharyngeal pumping (*Figure 5—figure supplement 1*), both of which are shown to be regulated by NPR-22 (*Figure 4*). Together with the finding that the *lury-1 (gk961835)* and *npr-22(ok1598)* deletion mutants showed similar egg-laying and feeding phenotypes (*Figure 6C–E*), we conclude that LURY-1 peptides act as endogenous ligands of NPR-22. The findings in our study and a previous study (*Palamiuc et al., 2017*) suggest that the two classes of NPR-22 ligands, LURY-1 peptides and FLP-7-derived peptides, mediate, at least in part, distinct functions.

Intriguingly, all the phenotypes caused by the multicopy expression of *lury-1(+)* (*Figure 4*) are strongly related to feeding. Suppression of feeding, facilitation of egg-laying, and attenuation of locomotory behavior are all induced by the presence of abundant food (*Waggoner et al., 2000*; *You et al., 2008*). Lifespan extension has also been closely associated with feeding (*Lakowski and Hekimi, 1998*). Our results that both the lifespan and feeding phenotypes of the *npr-22* mutants were rescued by the *npr-22* expression under the *ceh-19* promoter (*Figure 7A,E*) imply that the *lury-1(+)*-mediated lifespan extension was caused by reduced feeding. Moreover, the secretion of LURY-1 peptides apparently depends on the presence of food (*Figure 6A,B*). Taken together, these observations raise the possibility that LURY-1 peptides transmit food-related signals from pharynx, which plays critical roles in the *C. elegans* digestive system (*Albertson and Thomson, 1976*). Such a hypothesis is reminiscent of the mode of action of mammalian PYY and PP, which are also anorexigenic peptides derived from digestive organs (*Loh et al., 2015*). Interestingly, feeding suppressive effects of RYamide-containing peptides have also been observed in blowflies (*Maeda et al., 2015*) and kuruma shrimps (*Mekata et al., 2017*).

Although serotonin signaling in RIH is highly likely to mediate the effects of LURY-1 peptides on egg laying, the fact that serotonin signaling functions in HSN to facilitate egg-laying (*Chase and Koelle, 2007*) may complicate the interpretation of our results. The partial suppression of the LURY-1 peptide-mediated Egl-c phenotype by the *mod-5* mutations (*Figure 9C*) may imply the involvement of a cell(s) other than RIH and/or the parallel actions of LURY-1 peptides and serotonin. It is also unclear how NPR-22 controls the cell functions. Our results imply that NPR-22 suppresses the activity of the feeding pacemaker neurons MC and elevates serotonin signaling in RIH. NPR-22 may couple to different types of G-protein in different cells. Future studies will be needed to identify the downstream factors of NPR-22 and the functions of NPR-22-expressing cells other than MC and RIH.

It is noteworthy that the *lury-1*-expressing M1 and M2 neurons may have opposite functions in feeding. The M1 neuron responds to short wavelength light and stimulates spitting behavior (*Bhatla et al., 2015*), whereas the M2 neurons possess gap junction with the MC neurons and are stimulatory for pharyngeal pumping (*Albertson and Thomson, 1976*; *Trojanowski et al., 2014*). The significance of the expression of *lury-1* in these distinct neurons and the roles of the co-transmitters expressed in these neurons, such as acetylcholine and NLP/FLP neuropeptides (*Nathoo et al., 2001*; *Pereira et al., 2015*; *Rogers et al., 2003*), are important topics for future studies. We suppose that the *lury-1* promoter, which induces the highly specific transgene expression in M1 and M2 (*Figure 3A*), will be useful for examining the physiological functions of these neurons. Especially, manipulation of the neural activities or other cellular functions of these neurons will be important to precisely describe the roles of LURY-1 peptides in future studies, because we cannot exclude the possibility from our results that the multi-copy expression and/or the deletion of *lury-1* do not mimic the natural changes in the functions of LURY-1 peptides, and because the roles of the co-transmitters expressed in M1 and M2 remain unclear.

Biochemical identification and characterization of bioactive peptides have been relatively less reported in genetically-tractable small organisms such as *C. elegans*, making it difficult to conduct comparative analyses of peptides across classes or species. Our description of pharyngeal LURY-1 peptides in this study might lead to the discovery of relevant peptides and receptors in other organismal classes and re-evaluation of the functions of swallowing-related organs including pharynx, which have not received much attention in mammalian endocrinology, despite their direct involvement in feeding.

## Materials and methods

### Strains and culture

Bristol N2 was used as the wild-type *C. elegans*. Worms were raised on NGM using the standard methods (*Brenner, 1974*). The *Escherichia coli* strain HB101 was used as a food source. Strains used in this study are listed in *Supplementary file 2*.

### Construction of NPR-22-expressing cells

The full-length cDNA of *C. elegans* NPR-22a (*Y59H11AL.1*; GenBank accession number NM_001028369.2; residues 12–1343) was obtained by RT-PCR, with *C. elegans* cDNA as the template. The sense and antisense primers were 5′-cacccgtcatctaattcgtgagcaaa-3′ and 5′-tctatggtcttcta-tagctttccac-3′, respectively. The full-length cDNA of *C. elegans* NPR-22b (*Y59H11AL.1*; GenBank accession number NM_001028370.3; residues 35–1339) was obtained by gene synthesis (Eurofins Genomics, Tokyo, Japan). The amplified cDNA was cloned into a pcDNA3.2 vector (Invitrogen, Tokyo, Japan) and transfected into Chinese hamster ovary cells deficient of dihydrofolate reductase (CHO-dhfr-, provided by Dr. Kaoru Miyamoto and used as a standard cell line in the Kojima, Miyazato, and Ida laboratories). The CHO cell line was tested negative for mycoplasma contamination using VenorGem Classic Mycoplasma Testing PCR Kit (Minerva Biolabs, Berlin, Germany), and was not found to be on the list of commonly misidentified cell lines (International Cell Line Authentication Committee). Thereafter, stably expressing cells were selected using 1 mg/ml G418 (Nacalai Tesque, Kyoto, Japan). The selected cell lines, CHO-NPR-22a-line 7-3 and CHO-NPR-22b-line 11-2, which showed the highest expression of NPR-22a and NPR-22b mRNA, respectively, were used in this study.

### Purification of LURY-1-1 and LURY-1-2

During the purification process, the activity of LURY-1 peptides was followed by measuring changes in intracellular calcium concentrations ($[Ca^{2+}]_i$) with the FlexStation 3 fluorometric imaging plate reader (Molecular Devices, CA, USA) in a cell line stably expressing *Drosophila* CG5811 (CHO-CG5811-line 7-4), as described previously (*Ida et al., 2011*). Boiled frozen worms (100 g) were used as the starting material. A basic peptide fraction (SP-III) was prepared as described previously (*Ida et al., 2007*) and then fractionated on a Sephadex G-50 gel filtration column (1.7 × 130 cm; GE Healthcare, Tokyo, Japan). A portion (0.002%) of each fraction, equivalent to 2 mg frozen weight, was subjected to the assay using CHO-CG5811 cells. The active fractions were separated by CM-HPLC on a TSK CM-2SW column (4.6 × 250 mm; Tosoh, Tokyo, Japan) with a linear gradient of 0.01 to 0.6 M ammonium formate (pH6.5) in the presence of 10% acetonitrile (ACN) at a flow rate of 1 ml/min over 2 hr. Four distinct active fractions (fractions A–D, *Figure 1B*) were further purified by fractionation on the same column at pH 4.8. The active fractions were separated by RP-HPLC using a Symmetry C18 column (3.9 × 150 mm, Waters, MA, USA) with a linear gradient of 10% to 60% ACN containing 0.1% trifluoroacetic acid (TFA) at a flow rate of 1 ml/min for 80 min. The active peaks were finally purified manually by RP-HPLC using a Chemcosorb 3ODS-H column (2.1 × 75 mm; Chemco, Osaka, Japan) with a linear gradient of 10% to 60% ACN/0.1% TFA at a flow rate of 0.2 ml/min for 80 min. The final purified peptides were analyzed with a protein sequencer (model 494; Applied Biosystems, CA, USA) and a mass spectrometry (TOF/TOF 5800 system; AB Sciex, MA, USA).

### Cloning of prepro-LURY-1 cDNA

A tblastn search of the *C. elegans* nucleotide databases was performed using the amino-acid sequence of the purified peptides, and a *C. elegans* mRNA sequence (*lury-1/Y75B8A.11*; GenBank accession number NM_001268231.1) derived from an annotated nucleotide sequence was obtained. Based on this sequence, we designed sequence-specific primers (5′-aatccaatcatgctcacaagg-3′; 5′-gagaggtttcacaaaaagaacg-3′). RT-PCR was performed with *C. elegans* cDNA as the template. The candidate PCR product was subcloned into the pCR-II TOPO vector (Invitrogen) and sequenced by a DNA sequencer (model 3100; Applied Biosystems). The cDNA sequence was determined from six independent clones.

## Peptides

LURY-1-1, LURY-1-2, and FLP-7-3 were chemically synthesized by Scrum Inc. (Tokyo, Japan). FLP-1-6 ($PNFLRF-NH_2$), FLP-6-1 ($KSAYMRF-NH_2$), FLP-12 ($RNKFEFIRF-NH_2$), and FLP-22 ($SPSAKWMRF-NH_2$) were chemically synthesized by Peptide 2.0 Inc. (VA, USA).

## Fluorescence microscopy

Animals were mounted on 5% agar with 10 mM $NaN_3$. Soon after immobilization, images were captured using a Leica HCX PL APO 40×/0.85 CORR CS objective or an HC PL APO 10×/0.40 CS objective on a Leica TCS-SP5 confocal microscope. For examination of effects of starvation and refeeding (*Figure 6A,B* and *Figure 6—figure supplement 1B,C*), animals that were starved for two hours in basal buffer (50 mM NaCl, 5 mM potassium phosphate [pH 6.0], 1 mM $CaCl_2$, 1 mM $MgSO_4$, 0.05% gelatin) or refed on fresh HB101-seeded NGM plates again for the indicated times after the 2 hr starvation were observed.

## Egg-in-worm assays

The number of unlaid eggs in the uterus was quantified as described previously (*Koelle and Horvitz, 1996*). White-crescent stage L4 worms were transferred to fresh HB101-seeded NGM plates and were allowed to grow for 36–38 hr at 20℃. Synchronized adult worms were then individually dissolved in five-times diluted commercial bleach solution (Kitchen Heiter, Kao, Tokyo, Japan; final sodium hypochlorite concentration is ~1%) in a flat-bottom 96-well plate and their eggs were counted under a dissecting microscope. For examination of effects of starvation and refeeding (*Figures 6C* and *9B*), synchronized adult worms that were starved for two hours in basal buffer (50 mM NaCl, 5 mM potassium phosphate [pH 6.0], 1 mM $CaCl_2$, 1 mM $MgSO_4$, 0.05% gelatin) or refed on fresh HB101-seeded NGM plates again for the indicated times after the 2 hr starvation were dissolved.

## Egg-laying assays

White-crescent stage L4 worms were transferred to fresh HB101-seeded NGM plates and were allowed to grow for 36–38 hr at 20℃. The worms were again transferred to fresh HB101-seeded NGM plates gently and were allowed to lay eggs for 1 hr at 20℃. For examination of effects of starvation and refeeding (*Figure 6D*), synchronized adult worms were starved for two hours in basal buffer (50 mM NaCl, 5 mM potassium phosphate [pH 6.0], 1 mM $CaCl_2$, 1 mM $MgSO_4$, 0.05% gelatin) and refed on fresh HB101-seeded NGM plates for 30 min. Laid eggs were counted under a dissecting microscope.

## Pharyngeal pumping assays

Young adult worms were transferred to fresh HB101-seeded NGM plates and left for more than two hours at room temperature (23℃). Food pumping rate was then measured on the plates by counting grinder movements in the pharyngeal terminal bulb with a Zeiss (Oberkochen, Germany) Axiovert S100 inverted microscope. For examination of effects of starvation and refeeding (*Figure 6E*), worms that were starved for two hours in basal buffer (50 mM NaCl, 5 mM potassium phosphate [pH 6.0], 1 mM $CaCl_2$, 1 mM $MgSO_4$, 0.05% gelatin) were transferred to HB101-free NGM plates (time '0') or refed on fresh HB101-seeded NGM plates again for the indicated times (time '15', '30', '60', and '120') after the 2 hr starvation and their pumping rate was immediately measured.

## Lifespan assays

Lifespan assays were performed as previously described (*Ohno et al., 2014*) with modifications. Animals were raised at 20℃ until the L4 stage and then gently transferred to HB101-seeded plates containing 10 μg/ml 5-fluorodeoxyuridine (FUdR). Following transfer (day 0), the plates were incubated at 22℃. Animals were observed every other day and scored as dead when they no longer responded to gentle touch with a platinum wire.

## Roaming assays

Young adult worms were singly and gently transferred to fresh HB101-seeded NGM plates (diameter: 6 cm) and left for four hours in the dark at room temperature (23℃). After removing the worms,

each plate was superimposed on a grid containing 5 mm squares and the number of squares that were entered by the worm tracks was counted.

## Microinjection of synthetic peptides

Microinjection of peptides was performed as previously described (*Rogers et al., 2003*) with modifications. Synthetic LURY-1-1, LURY-1-2, or FLP-7-3 was diluted to a concentration of 10 µM in M9 buffer and then filtered through 0.22 µm filters (Merck Millipore, Billerica, MA). Each peptide solution or M9 buffer (control) was injected slightly posterior to the dorsal part of the pharyngeal terminal bulb using standard microinjection techniques (*Mello et al., 1991*). Each solution was injected until the diameter of its spread reached approximately 5–10 µm (one sixth to one third of the diameter of the terminal bulb of pharynx). The estimated injected amount is around 100 fL. The injected worms were transferred to fresh HB101-seeded NGM plates and incubated at room temperature (23°C). Their pharyngeal pumping rate and the number of eggs in their uterus were then counted 3 hr and 4 hr after microinjection, respectively.

## Germ-line transformation and integration of transgenes

Expression constructs were injected at 2–50 ng/µl along with a co-injection marker and pPD49.26 (a gift from A. Fire) as a carrier DNA. We utilized the co-injection marker transgenes *myo-3^prom^::venus*, *lin-44^prom^::gfp*, or *unc-122^prom^::mCherry* injected at 10–20 ng/µl. In each case, the total concentration of injected DNA was 100 ng/µl. In some strains (*Supplementary file 2*), the transgenes were amplified via PCR and the obtained linear vector-free DNA fragments were injected at 20–50 ng/µl after purification (*Etchberger and Hobert, 2008*). *lury-1(+)* genomic DNA (5,689 bp) was amplified via PCR using primers 5′-ctacagtaatcctaccgcactc-3′ and 5′-tattcaaatcacgggcggag-3′. For *peIs2413* and *peIs2414* strains, chromosomal integration of the *lury-1(+)* genomic DNA and the co-injected *myo-3^prom^::venus* transformation marker was induced using UV irradiation (300 µJ/cm$^2$). Each integrated strain was backcrossed six times with N2. *flp-7(+)* genomic DNA (4981 bp) was amplified via PCR using primers 5′-cactatgcggtcattacacgtc-3′ and 5′-aggcatcctctatcccaatataac-3′.

## Transgene constructions

The construction of pDEST-*venus*, pDEST-*mCherry*, pENTR-*ins-1^prom^*, pENTR-*ges-1^prom^*, pENTR-*myo-3^prom^*, pENTR-*rimb-1^prom^* (pENTR-H20p), pG-*myo-3^prom^::venus*, and pG-*unc-122^prom^::mCherry* was described previously (*Ohno et al., 2014*; *Tomioka et al., 2006*). For pDEST-*npr-22a* and pDEST-*npr-22b*, the *npr-22a* and *npr-22b* cDNAs, respectively, obtained by PCR from the NPR-22-pcDNA3.2 expression vectors (see above), were inserted into the *Nhe*I-*Kpn*I sites of the pPD-DEST vector. For pDEST-*FTF::npr-22a::SL2::mCherry*, the FRT-terminator (*let-858* 3′-UTR)-FRT fragment (FTF), *npr-22a* cDNA, the trans-spliced leader sequence SL2, and mCherry cDNA were concatenated and cloned into the pPD-DEST vector by a PCR-based method. For pDEST-FLP, the *Mlu*I-*Nhe*I fragment from pWD79-2RV (a gift from E. M. Jorgensen) was cloned into the pPD-DEST vector. The PCR-amplified *myo-2* promoter (1.2 kb) and *acr-2* promoter (1.9 kb) were cloned into pDONR201 (Invitrogen) through BP reaction (site-specific recombination) to create pENTR-*myo-2^prom^* and pENTR-*acr-2^prom^*, respectively. For pENTR-*lury-1^prom^*, pENTR-*npr-22^prom^*, pENTR-*flp-18^prom^*, pENTR-*flp-15^prom^*, pENTR-*ceh-19^prom^*, pENTR-*cat-1^prom^*, pENTR-*slt-1^prom^*, and pENTR-*acr-5^prom^*, the PCR-amplified *lury-1* promoter (3.5 kb), *npr-22* promoter (5.7 kb), *flp-18* promoter (4.2 kb), *flp-15* promoter (2.4 kb), *ceh-19* promoter (1.5 kb), *cat-1* promoter (4.0 kb), *slt-1* promoter (4.4 kb), and *acr-5* promoter (4.2 kb), respectively, were inserted into the pENTR1A vector (Invitrogen). The expression constructs of pG-*lury-1^prom^::venus*, pG-*lury-1^prom^::mCherry*, pG-*npr-22^prom^::venus*, pG-*npr-22^prom^::mCherry*, pG-*flp-18^prom^::venus*, pG-*flp-15^prom^::venus*, pG-*ceh-19^prom^::venus*, pG-*cat-1^prom^::venus*, pG-*slt-1^prom^::venus*, pG-*ins-1^prom^::venus*, pG-*acr-5^prom^::venus*, pG-*npr-22^prom^::npr-22a*, pG-*npr-22^prom^::npr-22b*, pG-*rimb-1^prom^::npr-22a*, pG-*ceh-19^prom^::npr-22a*, pG-*ins-1^prom^::npr-22a*, pG-*ges-1^prom^::npr-22a*, pG-*cat-1^prom^::npr-22a*, pG-*flp-15^prom^::npr-22a*, pG-*myo-2^prom^::npr-22a*, pG-*myo-3^prom^::npr-22a*, pG-*slt-1^prom^::npr-22a*, pG-*acr-2^prom^::npr-22a*, pG-*npr-22^prom^::FLP*, pG-*cat-1^prom^::FLP*, pG-*npr-22^prom^::FTF::npr-22a::SL2::mCherry*, and pG-*ceh-19^prom^::FTF::npr-22a::SL2::mCherry* were created by LR reactions (site-specific recombination) between the pENTR plasmids and the pDEST plasmids (*Ohno et al., 2014*). For pG-*lury-1^prom^::lury-1(with intron):venus*, the genomic coding region of *lury-1* (from the start codon to the last codon before the stop codon) was fused to *venus::*

*unc-54* 3'-UTR by a PCR-based method and cloned into the *Xho*I and *Apa*I sites of pG-*lury-1$^{prom}$::venus*. The expression constructs of *lury-1(+)* genomic DNA (5,689 bp, see above), *flp-7(+)* genomic DNA (4,981 bp, see above), *rimb-1$^{prom}$::lury-1(+)*, *glr-2$^{prom}$::venus* and *npr-1$^{prom}$::venus* were created by PCR-based methods. Further details of all constructs will be provided upon request.

## Data analyses

No statistical methods were used to predetermine sample size, although our sample sizes are similar to those previously reported. Most experiments were repeated on two to four separate days. Statistic analyses were performed using Prism v.5 (GraphPad software, San Diego, CA). We did not exclude any data in our statistic analyses. Data in all bar graphs are presented as mean ±S.E.M.

## Acknowledgements

We thank the Caenorhabditis Genetics Center and Mark Alkema for strains; Eri Iwamoto and Yoshiki Nakashima for technical assistance; Noboru Murakami, Pavak Shah, and members of the Ida and Iino laboratories for helpful comments and advice.

## Additional information

### Funding

| Funder | Grant reference number | Author |
| --- | --- | --- |
| Ministry of Education, Culture, Sports, Science, and Technology | Grants-in-aid (26450471) | Takanori Ida |
| Kato Memorial Bioscience Foundation | Research grant | Takanori Ida |
| Takeda Science Foundation | Research grant | Masayasu Kojima Takanori Ida |
| Ministry of Education, Culture, Sports, Science, and Technology | Grants-in-Aid for Innovative Areas 'Systems molecular ethology' (20115002) | Yuichi Iino |
| Ministry of Education, Culture, Sports, Science, and Technology | Grants-in-Aid for Innovative Areas 'Memory dynamism' (25115010) | Yuichi Iino |
| Ministry of Education, Culture, Sports, Science, and Technology | Grants-in-Aid for Innovative Areas 'Comprehensive Brain Science 774 Network' (221S0003) | Yuichi Iino |
| Ministry of Education, Culture, Sports, Science, and Technology | Grants-in-aid (26650025) | Masayasu Kojima |
| Ministry of Education, Culture, Sports, Science, and Technology | Grants-in-aid (16K07281) | Masayasu Kojima |
| Mishima Kaiun Memorial Foundation | Research grant | Hayao Ohno |
| Narishige Neuroscience Research Foundation | Research grant | Hayao Ohno |
| Human Frontier Science Program | Postdoctoral fellowship (LT000938/2017) | Hayao Ohno |

The funders had no role in study design, data collection and interpretation, or the decision to submit the work for publication.

## Author contributions
Hayao Ohno, Conceptualization, Data curation, Formal analysis, Funding acquisition, Investigation, Methodology, Writing—original draft, Writing—review and editing; Morikatsu Yoshida, Takahiro Sato, Mikiya Miyazato, Conceptualization, Data curation, Formal analysis, Investigation, Methodology, Writing—original draft, Writing—review and editing; Johji Kato, Conceptualization, Supervision, Writing—original draft, Writing—review and editing; Masayasu Kojima, Yuichi Iino, Conceptualization, Supervision, Funding acquisition, Writing—original draft, Writing—review and editing; Takanori Ida, Conceptualization, Data curation, Formal analysis, Funding acquisition, Investigation, Writing—original draft, Project administration, Writing—review and editing

## Author ORCIDs
Hayao Ohno (iD) http://orcid.org/0000-0003-2635-6964
Morikatsu Yoshida (iD) http://orcid.org/0000-0003-1577-1492
Takahiro Sato (iD) http://orcid.org/0000-0003-4364-2110
Takanori Ida (iD) http://orcid.org/0000-0003-3932-4611
Yuichi Iino (iD) http://orcid.org/0000-0002-0936-2660

## Decision letter and Author response
Decision letter https://doi.org/10.7554/eLife.28877.026
Author response https://doi.org/10.7554/eLife.28877.027

# Additional files
## Supplementary files
• Supplementary file 1. Results of lifespan analyses.
DOI: https://doi.org/10.7554/eLife.28877.023

• Supplementary file 2. Strains used in this study.
DOI: https://doi.org/10.7554/eLife.28877.024

• Transparent reporting form
DOI: https://doi.org/10.7554/eLife.28877.025

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
