## [Decision Letter]

Thank you for submitting your article "NPY-like peptides, CeRYamides, regulate food-evoked responses in *C. elegans*" for consideration by *eLife*. Your article has been reviewed by three peer reviewers, and the evaluation has been overseen by a Reviewing Editor and Eve Marder as the Senior Editor. The following individual involved in review of your submission has agreed to reveal his identity: Gáspár Jékely (Reviewer #3).

The reviewers have discussed the reviews with one another and the Reviewing Editor has drafted this decision to help you prepare a revised submission.

As you can see below, the two main issues raised by the reviewers:

1) Putting your work in broader context, particularly in relation to previous work in other invertebrate and vertebrate systems (reviewer #2). This includes a toning down of the novelty of your observations.

2) As detailed by reviewer #3, the phylogenetic analysis of the ligand/receptor system needs to be fundamentally revised and the system should be referred to as suggested by this reviewer.

We are including the full sets of initial reviews so that you see the context for 1) and 2) above.

Reviewer #1:

This manuscript reports the biochemical discovery of two *C. elegans* neuropeptides related to the RY-amide peptides of mammals and *Drosophila* whose release is regulated by feeding and that in turn regulate behaviors that respond to feeding. The authors identify the *C. elegans* NPR-22 protein as the receptor for the worm peptides using both cell-based activation assays and *C. elegans* genetic experiments. They show the peptides are expressed in and functionally released from a pair of neurons in the pharynx, the feeding organ of *C. elegans*. They show that transgenic overexpression or injection of the peptides into *C. elegans* affects four different behaviors that are normally regulated by feeding. They identify the specific neurons that expression NPR-22 and show which of these neurons appear to mediate each behavioral effect of the overexpressed peptides. Knocking out the peptide gene or the receptor gene has no effects on behavior under well fed conditions, but the authors show that two of the behavioral effects of re-feeding starved animals depend, albeit subtly, on both the peptide and receptor gene. Their results support a model in which re-feeding leads to high activity of the pharyngeal neurons, release of the peptides, and their subsequent control of specific behaviors.

This paper will be of interest to: 1) those studying neuropeptide signaling in the *C. elegans* system; 2) those studying NPY-like peptide signaling in mammals and fruit flies, and those interested in general in signaling control of feeding and control of other behaviors by feeding.

The paper has several strong points. The biochemical identification of active peptides from *C. elegans* extracts is very impressive. This has been rarely done in the past, and there is a huge problem in the field of trying to match orphan neuropeptide receptor homologs with ligands. It is impressive to see this biochemical approach used so successfully. Also, the experimental work in this paper is almost without exception meticulous and impressive. The supplemental figures, for example, show really impressive use of cell specific markers to identify cells expressing the peptide and receptor genes, and the quality of the work in these experiments typifies the rest of the work in the paper.

The identification of loss of function phenotypes for the peptide and receptor genes is a key issue in evaluating this paper. It is a general problem that loss of function mutant phenotypes for neuropeptides and their receptors are very difficult to identify. It appears that a typical neuropeptide is released and has effects only under very particular conditions and then has very specific effects, so finding these is like hunting for a needle in a haystack. The authors use a clever and potentially generalizable approach to solve this problem. They first identify the specific cells that express the peptide and receptor, and use the detailed existing knowledge of worm neural functions to be able to guess possible functions to test. They next find phenotypic effects of overexpressing or injecting the peptides. They then try varying the conditions in which they culture the worms (feeding and starving) to eventually find related phenotypic defects in the loss of function mutants. The paper could be criticized in that the loss of function effects seen are quite subtle, and certainly it would be nice if they were stronger. However, the current state of the field is such that it is impressive to see any loss of function effects for neuropeptide or neuropeptide receptor loss of function mutants.

Reviewer #2:

This is an interesting, detailed study which identifies, localizes and determines the feeding-related functions of a pair of RYamide neuropeptides in *C. elegans*. In addition to its contributions towards understanding the neuronal basis of behavior in *C. elegans*, this work establishes that this peptide family shares with family members in several other animals (molluscs, arthropods, mammals) a functional influence on feeding-related behaviors. Enthusiasm is a bit tempered by (a) some discomfort regarding the extent to which the conclusions are touted as establishing the universality of RYamides for feeding, and (b) confusion regarding the identity of FLP-7 and its receptor, NPR-22. These issues are discussed below. This work should be of interest to the broad readership of *eLife*.

1) Impact statement/Elsewhere - "unexpected parallels between the feeding regulatory mechanisms in *C. elegans*, arthropods, and mammals mediated by neuropeptides with common C-terminal RYamide motifs." At various locations in the manuscript (e.g. Abstract, Impact Statement, end of Introduction, Discussion), of which the partial quote from the Impact Statement is one example, the impact of the results are used to establish the generality of this peptide family with respect to feeding-related behaviors across animal classes. However, this "generality" was already evident. As indicated in the text (e.g. Introdusction and elsewhere), what this paper does (and does very nicely!) is to extend to *C. elegans* the results of many previous studies in a number of other, disparate animal groups (e.g. molluscs, insects, crustaceans, mammals). It seems more appropriate to indicate that these new data extend said generality to nematodes, rather than staking a claim that these new data establish that generality and that this outcome was unexpected.

2. FLP-7/NPR-22: I am confused regarding the identity and function(s) of the peptide FLP-7 and its receptor NPR-22. I had no such confusion upon reading the manuscript, but then out of curiosity I downloaded and read one of the references cited (Palamiuc et al., 2017). Please clarify whether the apparent peptide and receptor discrepancies between your manuscript and those cited below from Palamiuc et al., (2017) are my misunderstanding, or whether they are something that requires additional control experiments.

In Palamiuc et al., (2017), which is also a *C. elegans* study:

A. FLP-7 is stated unequivocally to be a tachykinin peptide, related to mammalian Substance P and sharing the same C-terminal amino acid sequence, and not a N-terminally extended RFamide peptide as is indicated in this manuscript. Its identification in Palamiuc et al., (2017) as a tachykinin instead of a RFamide peptide is troubling insofar as controls were performed with RFamide peptides, not tachykinins.

B. NPR-22 is stated unequivocally to be related to a mammalian tachykinin receptor (NKR). This discrepancy calls into question whether the NPR-22 deletion experiments represent exclusively complementary results to the CeRYamide manipulations or whether they also represent elimination of *C. elegans* tachykinin actions.

Reviewer #3:

In this paper Ohno and colleagues identify and characterise a new *C. elegans* RYamide neuropeptide. First, they perform biochemical purification and isolate *C. elegans* ligands for the *Drosophila* RYamide receptor, CG5811. Following fractionation and MS analysis, the authors identified RYamide peptides that derive from a hitherto undescribed *C. elegans* proneuropeptide. These RYamide peptides activated the *C. elegans* NPR-22 receptor, a homolog of *Drosophila* CG5811. The RYamide ligands were more potent agonists of NPR-22 than the previously identified FLP-7-3, an RFamide peptide.

Using available mutants and an impressive set of transgenic strains the authors then analysed in detail the function of CeRYa and NPR-22. They found that CeRYamide-1 and CeRYamide-2 act through NPR-22 to control multiple processes, including egg laying, pharyngeal pumping, and longevity.

The authors used transgenic rescue constructs restricted to individual neurons to reconstruct neuron-level RYamide signaling pathways.

They found that

1) CeRYamide signals through NPR-22

2) NPR-22 likely acts in the MC neuron to suppress food pumping

3) NPR-22 acts in RIH to facilitate egg-laying

4) CeRYamides facilitate egg-laying through serotonin in RIH

These are interesting results and demonstrate the power of *C. elegans* to reconstruct neuropeptide action at a single cell resolution.

I have some criticism regarding the phylogenetic analysis and the evolutionary interpretation of the results. First, the phylogenetic tree presented in Figure 1—figure supplement 1 is of insufficient quality. The taxon sampling and number of sequences is very low. The Neighbor Joining method used is not suitable to satisfactorily resolve GPCR phylogeny. Instead, the authors should use ML or Bayesian analysis with a larger number of sequences.

I performed a more extended analysis of these receptors. I could confirm that NPR-22 is orthologous to *Drosophila* CG5811, the RYamide receptor. However, this family is not orthologous to NPY receptors in vertebrates. The RYamide receptor family was previously shown to be related to luqin receptors from annelids and mollusks (PMID: 23637342, 26190115). These constitute a distinct family from NPY receptors, more closely related to leucokinin receptors.

Furthermore, analysis of the peptide precursor sequence also identifies Y75B8A.11 as a clear ortholog of luqin (also called abdominal ganglion neuropeptides L5-67) propeptides. These all have one or two RY/RFamide peptides directly after the signal peptide and a C-term domain with two conserved Cys residues. These are also present in Y75B8A.11.

Overall, the combined proneuropeptide and receptor evidence clearly shows that the authors studied the *C. elegans* luqin-luqin receptor system.

Luqin is an ancient bilaterian family and homologs can also be found in non-vertebrate deuterostomes (e.g. starfish PMID: 26865025, hemichordates), however, this family has been lost from vertebrates. An overview of luqin function in mollusks can be found in PMID: 25386166.

The authors should consider renaming the peptide to luqin, representing the more widely used name for this conserved family. The authors also need to rewrite the discussion and parts of the introduction. Especially, references to NPY signaling are less relevant, and the authors should compare their results to what is known about luqin signaling (not much). e.g., "Our results indicate that RYamide-containing peptides-NPY-like receptor axis is more widely" this is not true, since RYamides in insects are not orthologous to NPY.

In the introduction, the authors state that the comparative analyses of peptides across species have been hindered by their small size and low sequence conservation. This is not entirely true, there have been major developments in recent years in the comparative genomics of neuropeptides and their receptors, see: PMID: 23671109, 23637342, 25904544, 28444138, 26865025.

[Editors' note: further revisions were requested prior to acceptance, as described below.]

Thank you for resubmitting your work entitled "CeRYamides regulate food-evoked responses in *C. elegans*" for further consideration at *eLife*. Your revised article has been favorably evaluated by Eve Marder (Senior editor), a Reviewing editor, and two reviewers.

The manuscript has been much improved but there are some remaining issues that need to be addressed before acceptance, as outlined below. These are only editorial issues, but they are important. In particular, there was general agreement that the issue of homology assignment was not sufficiently addressed (point #1 including all the subpoints).

1) NPY peptides vs. Luqin peptides: It is essential that the author provide a proper description of evolutionary relationships.

a) The C-terminal identity is not a sufficient indicator of a family relationship, but the authors imply that it is. This has to be changed. A C-term RYamide motif can occur in several different, unrelated neuropeptides, including NPY, RYamide/luqin, repetitive FMRFamide, and occasionally in sulfakinine. Furthermore, not all NPY/NPF or luqin peptides end with an RYamide. Deuterostome luqins end with RWamide, and protostome NPFs are RFamides. see Figure S1D. in www.pnas.org/cgi/content/short/1221833110 for more details. Thus, the presence of an RYamide motif alone is not sufficient to establish the orthology of the precursors. It is therefore misleading to refer to a group of unrelated peptides as RYamides, since it does not necessarily mean a close relationship.

b) Abstract: "Both peptides are like luqin in invertebrates and contain C-terminal arginine-tyrosine-NH2 (RYamide) structures identical to those of the neuropeptide Y (NPY) family in vertebrates."

This needs to be changed. First, it would be good to state that both peptides derive from the same precursor. I suggest the following wording "Both peptides derive from the same precursor that is orthologous to invertebrate luqin/RYamide proneuropeptides." The second part of the sentence should be deleted, since the final two residues cannot be used as an evidence of orthology, but the wording implies a close relationship.

c) Last sentence of abstract: "Our results suggest that the roles of some RYamide-containing peptides and peptide-mediated negative feedback are widely conserved in feeding regulation among many animal classes." This needs to be changed. Since the exact relationship of NPY and luqin/RYamide is not clear and would require more extensive phylogenetic analysis, the authors also cannot argue about conservation. There are very few studies on luqins/RYamides, the only one I found is from the same group and shows that "When administered to blowflies, dRYamide-1 suppressed feeding motivation." Finding a function for an unstudied bilaterian neuropeptide family is interesting enough. The closing sentence should best just state what the authors found without overselling "Our results identified a critical role for luqin in feeding regulation..."

d) Impact statement: "feeding regulatory mechanisms in *C. elegans* and those in arthropods and mammals mediated by neuropeptides with common C-terminal RYamide motifs." This needs to be changed, mammals have no orthologs to luqin/RYamide

e) Introduction: "RYamides, peptides containing the C-terminal RYamide structure first discovered in the brachyuran crab Cancer borealis (Li et al., 2003), form a subgroup of luqins, and are also found in diverse invertebrates, such as crustaceans (Christie, 2014; Dircksen et al., 2011; Ma et al., 2010), mollusks (Proekt et al., 2005; Veenstra, 2010), and insects (Hauser et al., 2010; Ida et al., 2011; Roller et al., 2016)." This is not precise. RYamides do not form a subgroup of luqins, but are luqins that are named RYamides for historical reasons. The orthology of mollusk/annelid luqins and insect RYamides has only recently been recognised. These propeptides form one orthologous group, meaning that they trace back to one ancestral luqin peptide that was present in the common ancestor of all bilaterians. This peptide has been retained in ambulacrarians on the deuterostome side of the bilaterian tree, and lost from chordates. They have been retained in most protostomes. The history of the luqin/RYamide receptors exactly parallels this distribution, showing (as 30 other bilaterian peptide-receptor pairs) the long-term stability of peptide-receptor pairs in evolution. Members of this family should best be called luqin, to avoid confusion. The name RYamide is preferred by the authors, probably because they want to emphasise its similarity to vertebrate NPY, however, this is now known, that luqin/NPY systems are not orthologous. In the text, the authors should also mention that luqin has been identified in *S. purpuratus* and *S. kowalevskii* (two deuterosomes), but has been lost from chordates.

f) The tree in (Figure 1—figure supplement 1) luqin and RYamide receptors are highlighted in two colours. This is unfortunate, since it is one orthologous family, and that part of the tree only represents the species tree. The two sides of the orthology group (with lophotrochozoans and ecdysozoans on the two sides) should be highlighted in the same colour and referred to as luqin/RYamide receptors.

g) There is insufficient relationship information between NPY-family receptors and luqin receptors. The exact paralogy relationships of luqin receptors and the other families (tachykinin, NPY, leucokinin) are not resolved by the tree shown, so this tree cannot be used to suggest a more direct paralogy relationship between luqin and NPY receptors. However, the authors could discuss that luqin, NPY, leucokinin, and tachykinin families are distantly related, but their relationships have not yet been clearly resolved, as described in these papers:

Figure 2 in www.pnas.org/cgi/doi/10.1073/pnas.1219956110 and Figure 3 in www.pnas.org/cgi/doi/10.1073/pnas.1221833110

h) Nomenclature: To use Ce in the peptide name is not very fortunate, because the same peptide would then be called CbRY in *C. briggsae* and so on. This propeptide is a *C. elegans* luqin, and the receptor is a *C. elegans* luqin receptor. Furthermore, the propeptide that the authors identified has already been predicted as a *C. elegans* luqin in the very elegant bioinformatic study of Oliver Mirabeau in 2013 (www.pnas.org/cgi/doi/10.1073/pnas.1219956110). This also shows that the identity of CeRY as a luqin is very clear. See Mirabeau & Joly Dataset S1, alignment #30 www.pnas.org/cgi/doi/10.1073/pnas.1219956110 a quote from their paper: "We give predictions for *C. elegans* peptides that had not been characterized to date, which include TK, DH31/Calc, SIFa, LK, luqin, and AstA,B,and C (Dataset S1)."

i) Results section: "We designated these PALLSRY-NH2 and AVLPRY-NH2 peptides as CeRYamide-1 and CeRYamide-2, respectively. The proneuropeptide has a characteristic of luqins with canonical pair of cysteines in the C-terminal portion (Figure 2)." The peptides should be referred to as luqin-1 and luqin-2 (or, to accommodate the 4 letter restriction of gene names in *C. elegans*, luq or luqi).

2. Co-transmitters: In the Authors’ Reply to Reviewer Comments, they indicate that there are identified co-transmitters (including other neuropeptides) in the CeRYamide-containing M1 and M2 neurons but, given the data in this manuscript, the CeRYamide peptides are clearly the most important with respect to the studied behaviors. This may be true, but in fact that conclusion is premature. While we do agree that this conclusion may be correct, unless one actually performs experiments to understand the role(s) of the co-transmitters, it is simply not possible to know whether (a) the CeRYamides are indeed predominant in these behaviors, or (b) the imbalance between the actions of CeRyamides vs. its co-transmitters created by the manipulations in this manuscript provides heightened or diminished actions of one or more co-transmitter that are pivotal to the behavioral changes observed. Without a doubt, the CeRYamides play an important role in the studied behaviors, but considering that role to be pre-eminent seems premature. This point is also further addressed in the next comment (Major Comment #3).

A) Insofar as it is suggested in the Discussion section that the two CeRyamide-containing neurons (M1, M2) may have distinct functions, it seems relevant to include in that discussion the identity of the co-transmitters in those neurons since they may well contribute to the distinct functions.

3. Whole animal manipulations vs. individual neuron functions (Discussion section): The fact that the two CeRYamide-containing neurons (M1, M2) "may have opposite functions in feeding" gives one pause with respect to how reliably the experiments performed in this paper mimic the natural functions of neuronally-released CeRYamides. By way of example from a different model system, physiological studies in the crustacean stomatogastric ganglion have shown that directly applying an identified peptide (proctolin) to this ganglion does not mimic the actions of 2 of the 3 types of proctolin-containing neurons that innervate this ganglion (reviewed in: Nusbaum et al., 2001; 2017). Whole system manipulations of a neuropeptide, as performed in the current manuscript, provide considerable valuable information regarding the action(s) of the manipulated peptide (as do the results in this paper), but those actions should never be considered to be identical to those of the peptide when they are neuronally released, unless this relationship is established by experiments. There are numerous reasons for the likelihood of discrepancies between these two approaches (e.g. see Nusbaum et al., 2017), one of which is the aforementioned (comment #3) presence of co-transmitters. Therefore, it is prudent to provide in the Discussion section an explicit indication that future studies will determine the extent to which the behavioral consequences established in this paper reflect the actions of neuronally-released CeRYamide.

---

## [Author Response]

*Reviewer #1:*

*[…] The identification of loss of function phenotypes for the peptide and receptor genes is a key issue in evaluating this paper. It is a general problem that loss of function mutant phenotypes for neuropeptides and their receptors are very difficult to identify. It appears that a typical neuropeptide is released and has effects only under very particular conditions and then has very specific effects, so finding these is like hunting for a needle in a haystack. The authors use a clever and potentially generalizable approach to solve this problem. They first identify the specific cells that express the peptide and receptor, and use the detailed existing knowledge of worm neural functions to be able to guess possible functions to test. They next find phenotypic effects of overexpressing or injecting the peptides. They then try varying the conditions in which they culture the worms (feeding and starving) to eventually find related phenotypic defects in the loss of function mutants. The paper could be criticized in that the loss of function effects seen are quite subtle, and certainly it would be nice if they were stronger. However, the current state of the field is such that it is impressive to see any loss of function effects for neuropeptide or neuropeptide receptor loss of function mutants.*

We sincerely appreciate the thoughtful comments and evaluation by the reviewer. As the reviewer pointed out, we wished that the loss-of-function phenotypes were stronger, but the defect was towards the direction opposite to the overexpression phenotype, and manifested under particular conditions. This is exactly how we reasoned and found the phenotype.

*Reviewer #2:*

*This is an interesting, detailed study which identifies, localizes and determines the feeding-related functions of a pair of RYamide neuropeptides in C. elegans. In addition to its contributions towards understanding the neuronal basis of behavior in C. elegans, this work establishes that this peptide family shares with family members in several other animals (molluscs, arthropods, mammals) a functional influence on feeding-related behaviors. Enthusiasm is a bit tempered by (a) some discomfort regarding the extent to which the conclusions are touted as establishing the universality of RYamides for feeding, and (b) confusion regarding the identity of FLP-7 and its receptor, NPR-22. These issues are discussed below. This work should be of interest to the broad readership of eLife.*

We are grateful to the reviewer for evaluating the overall significance of the work.

*1) Impact statement/Elsewhere - "unexpected parallels between the feeding regulatory mechanisms in C. elegans, arthropods, and mammals mediated by neuropeptides with common C-terminal RYamide motifs." At various locations in the manuscript (e.g. Abstract, Impact Statement, end of Introduction, Discussion), of which the partial quote from the Impact Statement is one example, the impact of the results are used to establish the generality of this peptide family with respect to feeding-related behaviors across animal classes. However, this "generality" was already evident. As indicated in the text (e.g. Introduction and elsewhere), what this paper does (and does very nicely!) is to extend to C. elegans the results of many previous studies in a number of other, disparate animal groups (e.g. molluscs, insects, crustaceans, mammals). It seems more appropriate to indicate that these new data extend said generality to nematodes, rather than staking a claim that these new data establish that generality and that this outcome was unexpected.*

We are sorry about our ignorance of the previous understanding in literature, and thank the reviewer for pointing this out. As suggested, we changed the statements as follows:

In Impact Statement, we deleted "unexpected", leaving the following statement: "CeRYamide peptides reveal parallels between the feeding regulatory mechanisms in *C. elegans* and those in arthropods and mammals".

In Introduction, we deleted "unexpected" and rephrased the sentence, leaving the following statement: "Here, we demonstrate that the RYamide-containing peptides are conserved in *C. elegans* […]".

*2. FLP-7/NPR-22: I am confused regarding the identity and function(s) of the peptide FLP-7 and its receptor NPR-22. I had no such confusion upon reading the manuscript, but then out of curiosity I downloaded and read one of the references cited (Palamiuc et al., 2017). Please clarify whether the apparent peptide and receptor discrepancies between your manuscript and those cited below from Palamiuc et al., (2017) are my misunderstanding, or whether they are something that requires additional control experiments.*

*In Palamiuc et al., (2017), which is also a C. elegans study:*

*A. FLP-7 is stated unequivocally to be a tachykinin peptide, related to mammalian Substance P and sharing the same C-terminal amino acid sequence, and not a N-terminally extended RFamide peptide as is indicated in this manuscript. Its identification in Palamiuc et al., (2017) as a tachykinin instead of a RFamide peptide is troubling insofar as controls were performed with RFamide peptides, not tachykinins.*

*B. NPR-22 is stated unequivocally to be related to a mammalian tachykinin receptor (NKR). This discrepancy calls into question whether the NPR-22 deletion experiments represent exclusively complementary results to the CeRYamide manipulations or whether they also represent elimination of C. elegans tachykinin actions.*

As pointed out by the reviewer, Palamiuc et al., claimed that FLP-7 is a ligand of NPR-22 based on the previous cell-line assay (Mertens et al., 2006, *Peptides*). This is consistent with our assay and our results show that CeRYamides are more potent agonists of NPR-22 (Figure 2). Palamiuc et al., also showed that loss-of-function mutants of *flp-7* and *npr-22* show a similar defect in fat reduction in the presence of serotonin, consistent with the possibility that FLP-7 acts on NPR-22 in vivo. This is not inconsistent with our result that CeRYamides are ligands for NPR-22, and it is possible that both of these different peptides act on NPR-22. We added discussion on this possibility in the text.

As discussed by Reviewer #3, their assignment of FLP-7 to tachykinin and NPR-22 to tachykinin receptor is not based on strong evidence. In Palamiuc et al., (2017), the sequences of FLP-7-1 and Substance P were shown as SPMQRSSMVRFGKR and RPKPQQFFGLMGKR, respectively (Supplementary Figure 1). However, the last two basic amino acid residues (KR) are the putative cleavage site in the precursors and the third residue from the last (Glycine) serves as the putative amide donor for C-terminal amidation. Thus, the last three residues (GKR) are a fairly common motif in peptide precursors and FLP-7 peptides are highly likely to be RFamides. Considering this, in our opinion, the similarity in the sequences of mature FLP-7-1 and substance P, SPMQRSSMVRFamide and RPKPQQFFGLMamide, are quite limited.

*Reviewer #3:*

*[…] I have some criticism regarding the phylogenetic analysis and the evolutionary interpretation of the results. First, the phylogenetic tree presented in Figure 1—figure supplement 1 is of insufficient quality. The taxon sampling and number of sequences is very low. The Neighbor Joining method used is not suitable to satisfactorily resolve GPCR phylogeny. Instead, the authors should use ML or Bayesian analysis with a larger number of sequences.*

*I performed a more extended analysis of these receptors. I could confirm that NPR-22 is orthologous to Drosophila CG5811, the RYamide receptor. However, this family is not orthologous to NPY receptors in vertebrates. The RYamide receptor family was previously shown to be related to luqin receptors from annelids and mollusks (PMID: 23637342, 26190115). These constitute a distinct family from NPY receptors, more closely related to leucokinin receptors.*

We thank the reviewer for raising this important issue. As suggested, we also performed phylogenetic analysis of the receptor with increased number of receptors from various organisms. As already shown by the reviewer, NPR-22 belongs to the Luqin receptor/RYamide receptor group, which is distantly related to NPY receptor family and tachykinin receptor family. Figure 1—figure supplement 1 was replaced with this revised version. Also, throughout the text we indicated that NPR-22 belongs to the luqin/RYamide receptor family.

*Furthermore, analysis of the peptide precursor sequence also identifies Y75B8A.11 as a clear ortholog of luqin (also called abdominal ganglion neuropeptides L5-67) propeptides. These all have one or two RY/RFamide peptides directly after the signal peptide and a C-term domain with two conserved Cys residues. These are also present in Y75B8A.11.*

*Overall, the combined proneuropeptide and receptor evidence clearly shows that the authors studied the C. elegans luqin-luqin receptor system.*

*Luqin is an ancient bilaterian family and homologs can also be found in non-vertebrate deuterostomes (e.g. starfish PMID: 26865025, hemichordates), however, this family has been lost from vertebrates. An overview of luqin function in mollusks can be found in PMID: 25386166.*

*The authors should consider renaming the peptide to luqin, representing the more widely used name for this conserved family. The authors also need to rewrite the discussion and parts of the introduction. Especially, references to NPY signaling are less relevant, and the authors should compare their results to what is known about luqin signaling (not much).*

We thank also for leading us to these literature. Based on the alignments made in previous studies, RYamides form a subgroup of the luqin family, so we keep the nomenclature CeRYamide but clearly indicated in the revised manuscript that the precursor is like members of the luqin family.

*E.g., "Our results indicate that RYamide-containing peptides-NPY-like receptor axis is more widely" this is not true, since RYamides in insects are not orthologous to NPY.*

We corrected such sentences because of the distant relationship of CeRYamide and its receptor to NPY. The "NPY-like" was omitted from the title. "axis of RYamide-containing peptides and NPY-like receptor is unexpectedly conserved in *C. elegans*" was changed to "RYamide-containing peptides are conserved in *C. elegans*". "Our results indicate that RYamide-containing peptides–NPY-like receptor axis is more widely conserved among animal classes than previously thought" was deleted.

Although with the distant relationship, we consider introduction of NPY family is beneficial because peptidergic control of feeding-related functions is a general scheme. So the introduction now refers to all relevant information including NPY and luqin.

*In the introduction, the authors state that the comparative analyses of peptides across species have been hindered by their small size and low sequence conservation. This is not entirely true, there have been major developments in recent years in the comparative genomics of neuropeptides and their receptors, see: PMID: 23671109, 23637342, 25904544, 28444138, 26865025.*

Thank you again for the guidance. We cited these references.

[Editors' note: further revisions were requested prior to acceptance, as described below.]

*1) NPY peptides vs. Luqin peptides: It is essential that the author provide a proper description of evolutionary relationships.*

*a) The C-terminal identity is not a sufficient indicator of a family relationship, but the authors imply that it is. This has to be changed. A C-term RYamide motif can occur in several different, unrelated neuropeptides, including NPY, RYamide/luqin, repetitive FMRFamide, and occasionally in sulfakinine. Furthermore, not all NPY/NPF or luqin peptides end with an RYamide. Deuterostome luqins end with RWamide, and protostome NPFs are RFamides. see Figure S1D. in www.pnas.org/cgi/content/short/1221833110 for more details.*

*Thus, the presence of an RYamide motif alone is not sufficient to establish the orthology of the precursors. It is therefore misleading to refer to a group of unrelated peptides as RYamides, since it does not necessarily mean a close relationship*

As the reviewer suggested, we revised Introduction and Discussion, and removed indeliberate reference to all RYamide-containing peptides as “RYamides”. For example, we changed the phrase “RYamides, peptides containing the C-terminal RYamide structure first discovered in the brachyuran crab *Cancer borealis* (Li et al., 2003),” to “In invertebrates, peptides containing the C-terminal RYamide structure were first discovered in the brachyuran crab *Cancer borealis* (Li et al., 2003)”. The term "RYamides" is only used for putative luqin orthologues based on the related structures of the propeptides in the revised manuscript, as described further below.

*b) Abstract: "Both peptides are like luqin in invertebrates and contain C-terminal arginine-tyrosine-NH2 (RYamide) structures identical to those of the neuropeptide Y (NPY) family in vertebrates." This needs to be changed. First, it would be good to state that both peptides derive from the same precursor. I suggest the following wording "Both peptides derive from the same precursor that is orthologous to invertebrate luqin/RYamide proneuropeptides." The second part of the sentence should be deleted, since the final two residues cannot be used as an evidence of orthology, but the wording implies a close relationship.*

As suggested, we changed the sentence to “Both peptides derive from the same precursor that is orthologous to invertebrate luqin/arginine-tyrosine-NH_2_ (RYamide) proneuropeptides.”

*c) Last sentence of abstract: "Our results suggest that the roles of some RYamide-containing peptides and peptide-mediated negative feedback are widely conserved in feeding regulation among many animal classes." This needs to be changed. Since the exact relationship of NPY and luqin/RYamide is not clear and would require more extensive phylogenetic analysis, the authors also cannot argue about conservation. There are very few studies on luqins/RYamides, the only one I found is from the same group and shows that "When administered to blowflies, dRYamide-1 suppressed feeding motivation." Finding a function for an unstudied bilaterian neuropeptide family is interesting enough. The closing sentence should best just state what the authors found without overselling "Our results identified a critical role for luqin in feeding regulation"*

We omitted reference to NPY and changed the sentence to “Our results identified a critical role for luqin-like RYamides in feeding-related processes and suggested that peptide-mediated negative feedback is important for satiety regulation in *C. elegans*”, to avoid any mention about conservation among species.

*d) Impact statement: "feeding regulatory mechanisms in C. elegans and those in arthropods and mammals mediated by neuropeptides with common C-terminal RYamide motifs." This needs to be changed, mammals have no orthologs to luqin/RYamide*

We changed the sentence to “Identification and functional characterization of *C. elegans* luqin-like arginine-tyrosine-NH_2_ (RYamide) peptides reveal their critical role in feeding-related processes.”

*e) Introduction: "RYamides, peptides containing the C-terminal RYamide structure first discovered in the brachyuran crab Cancer borealis (Li et al., 2003), form a subgroup of luqins, and are also found in diverse invertebrates, such as crustaceans (Christie, 2014; Dircksen et al., 2011; Ma et al., 2010), mollusks (Proekt et al., 2005; Veenstra, 2010), and insects (Hauser et al., 2010; Ida et al., 2011; Roller et al., 2016)." This is not precise. RYamides do not form a subgroup of luqins, but are luqins that are named RYamides for historical reasons. The orthology of mollusk/annelid luqins and insect RYamides has only recently been recognised. These propeptides form one orthologous group, meaning that they trace back to one ancestral luqin peptide that was present in the common ancestor of all bilaterians. This peptide has been retained in ambulacrarians on the deuterostome side of the bilaterian tree, and lost from chordates. They have been retained in most protostomes. The history of the luqin/RYamide receptors exactly parallels this distribution, showing (as 30 other bilaterian peptide-receptor pairs) the long-term stability of peptide-receptor pairs in evolution. Members of this family should best be called luqin, to avoid confusion. The name RYamide is preferred by the authors, probably because they want to emphasise its similarity to vertebrate NPY, however, this is now known, that luqin/NPY systems are not orthologous. In the text, the authors should also mention that luqin has been identified in S. purpuratus and S. kowalevskii (two deuterosomes), but has been lost from chordates.*

As suggested, we deleted the phrase “(RYamides) form a subgroup of luqins”. We also added in introduction the following: “Orthologs of luqin/RYamides precursors have been also found in deuterostomes (*S. purpuratus* and *S. kowalevskii*), but not in chordates (Jékely, 2013; Mirabeau and Joly, 2013).”

In the revised manuscript we called the two *C. elegans* peptides “luqin-like RYamide peptides”, but not “luqins”, because of the following reasons:

Many ecdysozoan peptides have been called “RYamide” in previous studies. Mirabeau and Joly (2013) also used the expression “alignment of lophotrochozoan luqin and arthropod RYamide peptides” (Dataset S1, #30).

The presence of the paired-cysteine motif is apparently not a perfect hallmark of the luqin/RYamide family. The original dataset of Mirabeau and Joly (2013) includes 13 members of the luqin/RYamide (http://neuroevo.org), but out of them three members (*jgi_213930_Ctel, gi_18866372_Agam*, and *Ensembl_FBtr0310523_NepY_Dmel*) were not presented in the alignment in their paper (Dataset S1, #30). Of these, *jgi_213930_Ctel* and *Ensembl_FBtr0310523_NepY_Dmel* do not have the paired-cysteine motif in the C-terminal portion. (And *gi_18866372_Agam* has a long signal peptide.) In addition, the conserved amino acids (S/T)-G and (V/I/L)-(P/Y) between two cysteines are not present in *C. elegans*.

Although there are exceptions, lophotrochozoan luqins and ecdysozoan RYamides have their own characteristics: (a) One peptide is produced from a lophotrochozoan luqin precursor, but two peptides are produced from an ecdysozoan RYamide precursor and (b) The C-terminal structure of most ecdysozoan RYamide peptides is RYamide. We surely think that lophotrochozoan luqins and ecdysozoan RYamides evolved from a common ancestor as the reviewers pointed out, but the *C. elegans* peptides apparently have RYamide-like features.

it is currently unclear whether lophotrochozoan luqins and ecdysozoan RYamides are functionally relevant, because the physiological functions of these peptides, especially of luqins, have been largely unexplored.

*f) The tree in (Figure 1—figure supplement 1) luqin and RYamide receptors are highlighted in two colours. This is unfortunate, since it is one orthologous family, and that part of the tree only represents the species tree. The two sides of the orthology group (with lophotrochozoans and ecdysozoans on the two sides) should be highlighted in the same colour and referred to as luqin/RYamide receptors.*

We refined the phylogenetic tree (*Figure 1—figure supplement 1*) as suggested.

*g) There is insufficient relationship information between NPY-family receptors and luqin receptors. The exact paralogy relationships of luqin receptors and the other families (tachykinin, NPY, leucokinin) are not resolved by the tree shown, so this tree cannot be used to suggest a more direct paralogy relationship between luqin and NPY receptors. However, the authors could discuss that luqin, NPY, leucokinin, and tachykinin families are distantly related, but there relationships have not yet been clearly resolved, as described in these papers:*

*Figure 2 in www.pnas.org/cgi/doi/10.1073/pnas.1219956110 and Figure 3 in www.pnas.org/cgi/doi/10.1073/pnas.1221833110*

As suggested, we added in Introduction the following: “This phylogenic analysis shows that NPR-22 and CG5811 belong to the luqin/RYamide receptor family and may be distantly related to tachykinin, NPY, and leucokinin receptors (Figure 1—figure supplement 1), although the evolutionary relationship between the luqin/RYamide receptor group and other receptor groups has not so far been clear (Jékely, 2013; Mirabeau and Joly, 2013)

*h) Nomenclature: To use Ce in the peptide name is not very fortunate, because the same peptide would then be called CbRY in C. briggsae and so on. This propeptide is a C. elegans luqin, and the receptor is a C. elegans luqin receptor. Furthermore, the propeptide that the authors identified has already been predicted as a C. elegans luqin in the very elegant bioinformatic study of Oliver Mirabeau in 2013 (www.pnas.org/cgi/doi/10.1073/pnas.1219956110). This also shows that the identity of CeRY as a luqin is very clear. See Mirabeau & Joly Dataset S1, alignment #30 www.pnas.org/cgi/doi/10.1073/pnas.1219956110 a quote from their paper: "We give predictions for C. elegans peptides that had not been characterized to date, which include TK, DH31/Calc, SIFa, LK, luqin, and AstA,B,and C (Dataset S1)."*

We discussed the gene name with the WormBase staff and assigned *Y75B8A.11* the gene name *lury-1* (LUqin-like RYamide peptides or LUqin/RYamide peptides). According to the new gene name and the *C. elegans* peptide nomenclature (Li and Kim, 2008, Wormbook), we changed the peptide names “CeRYamide-1 and -2” to “LURY-1-1 and -2”.

We also added the following explanation in Results, “These two peptides were encoded by the same gene, *Y75B8A.11*, which was predicted as an ortholog of luqin/RYamide precursors in a bioinformatic study (Mirabeau and Joly, 2013)

*A) Results section: "We designated these PALLSRY-NH2 and AVLPRY-NH2 peptides as CeRYamide-1 and CeRYamide-2, respectively. The proneuropeptide has a characteristic of luqins with canonical pair of cysteines in the C-terminal portion (Figure 2)." The peptides should be referred to as luqin-1 and luqin-2 (or, to accommodate the 4 letter restriction of gene names in C.elegans, luq or luqi)*

As noted above, we named the gene *lury-1* and changed the peptide names accordingly.

*2) Co-transmitters: In the Authors’ Reply to Reviewer Comments, they indicate that there are identified co-transmitters (including other neuropeptides) in the CeRYamide-containing M1 and M2 neurons but, given the data in this manuscript, the CeRYamide peptides are clearly the most important with respect to the studied behaviors. This may be true, but in fact that conclusion is premature. While we do agree that this conclusion may be correct, unless one actually performs experiments to understand the role(s) of the co-transmitters, it is simply not possible to know whether (a) the CeRYamides are indeed predominant in these behaviors, or (b) the imbalance between the actions of CeRyamides vs. its co-transmitters created by the manipulations in this manuscript provides heightened or diminished actions of one or more co-transmitter that are pivotal to the behavioral changes observed. Without a doubt, the CeRYamides play an important role in the studied behaviors, but considering that role to be pre-eminent seems premature. This point is also further addressed in the next comment (Major Comment #3).*

It is surely possible that the multi-copy expression and/or the deletion of the peptides affect the functions of the co-transmitters, although we do not think that the synthetic peptides administered into the worm body (Figure 5) act specifically on a co-transmitter(s) expressed in M1 and M2. We added statements about the co-transmitters in Discussion (see below).

*(A) Insofar as it is suggested in the Discussion section that the two CeRyamide-containing neurons (M1, M2) may have distinct functions, it seems relevant to include in that discussion the identity of the co-transmitters in those neurons since they may well contribute to the distinct functions.*

As suggested, we added the following explanation in Discussion: “The significance of the expression of *lury-1* in these distinct neurons and the roles of the co-transmitters expressed in these neurons, such as acetylcholine and NLP/FLP neuropeptides (Nathoo et al., 2001; Pereira et al., 2015; Rogers et al., 2003), are important topics for future studies,” and “the roles of the co-transmitters expressed in M1 and M2 remain unclear

*3. Whole animal manipulations vs. individual neuron functions (Discussion section): The fact that the two CeRYamide-containing neurons (M1, M2) "may have opposite functions in feeding" gives one pause with respect to how reliably the experiments performed in this paper mimic the natural functions of neuronally-released CeRYamides. By way of example from a different model system, physiological studies in the crustacean stomatogastric ganglion have shown that directly applying an identified peptide (proctolin) to this ganglion does not mimic the actions of 2 of the 3 types of proctolin-containing neurons that innervate this ganglion (reviewed in: Nusbaum et al., 2001; 2017). Whole system manipulations of a neuropeptide, as performed in the current manuscript, provide considerable valuable information regarding the action(s) of the manipulated peptide (as do the results in this paper), but those actions should never be considered to be identical to those of the peptide when they are neuronally released, unless this relationship is established by experiments. There are numerous reasons for the likelihood of discrepancies between these two approaches (e.g. see Nusbaum et al., 2017), one of which is the aforementioned (comment #3) presence of co-transmitters. Therefore, it is prudent to provide in the Discussion section an explicit indication that future studies will determine the extent to which the behavioral consequences established in this paper reflect the actions of neuronally-released CeRYamide.*

We added the following sentence in Discussion section: “Especially, manipulation of the neural activities or other cellular functions of these neurons will be important to precisely describe the roles of LURY-1 peptides in future studies, because we cannot exclude the possibility from our results that the multi-copy expression and/or the deletion of *lury-1* do not mimic the natural changes in the functions of LURY-1 peptides, and because the roles of the co-transmitters expressed in M1 and M2 remain unclear.”